# Impact of protein and small molecule interactions on kinase conformations

**Valentina Kugler[1,2†], Selina Schwaighofer[1,2†], Andreas Feichtner[1,2], Florian Enzler[3], Jakob Fleischmann[1,2], Sophie Strich[1,2], Sarah Schwarz[2], Rebecca Wilson[4], Philipp Tschaikner[2,5], Jakob Troppmair[3], Veronika Sexl[6], Pascal Meier[4], Teresa Kaserer[7], Eduard Stefan[1,2,5]\***

[1]Institute for Molecular Biology and Center for Molecular Biosciences Innsbruck (CMBI), University of Innsbruck, Innsbruck, Austria; [2]Tyrolean Cancer Research Institute (TKFI), Innsbruck, Austria; [3]Daniel Swarovski Research Laboratory, Department of Visceral, Transplant and Thoracic Surgery, Medical University of Innsbruck, Innsbruck, Austria; [4]The Breast Cancer Now Toby Robins Research Centre, The Institute of Cancer Research, London, United Kingdom; [5]KinCon biolabs GmbH, Innsbruck, Austria; [6]University of Innsbruck, Innsbruck, Austria; [7]Institute of Pharmacy/Pharmaceutical Chemistry and Center for Molecular Biosciences Innsbruck (CMBI), University of Innsbruck, Innsbruck, Austria

**\*For correspondence:**
Eduard.Stefan@uibk.ac.at

[†]These authors contributed equally to this work

**Abstract** Protein kinases act as central molecular switches in the control of cellular functions. Alterations in the regulation and function of protein kinases may provoke diseases including cancer. In this study we investigate the conformational states of such disease-associated kinases using the high sensitivity of the kinase conformation (KinCon) reporter system. We first track BRAF kinase activity conformational changes upon melanoma drug binding. Second, we also use the KinCon reporter technology to examine the impact of regulatory protein interactions on LKB1 kinase tumor suppressor functions. Third, we explore the conformational dynamics of RIP kinases in response to TNF pathway activation and small molecule interactions. Finally, we show that CDK4/6 interactions with regulatory proteins alter conformations which remain unaffected in the presence of clinically applied inhibitors. Apart from its predictive value, the KinCon technology helps to identify cellular factors that impact drug efficacies. The understanding of the structural dynamics of full-length protein kinases when interacting with small molecule inhibitors or regulatory proteins is crucial for designing more effective therapeutic strategies.

## eLife assessment

This article reports an **important** bioluminescence-based reporter system to evaluate kinase conformations. This assay is applied to four different kinases that have unique, very special regulatory features, thereby indicating that the assay can be used to provide **convincing** evidence on the conformational state of a large number of kinases. This paper will be of interest to researchers working on kinases and their conformational states.

## Introduction

The human kinome is encoded by more than 500 genes which lead to the synthesis of functionally diverse kinases. Kinases are different in their domain composition but all contain a structurally conserved phosphotransferase domain (*Manning, 2009*, *Kornev and Taylor, 2010*). Conventionally, protein kinases function as enzymes that enable the transfer of a phosphate group from ATP to

defined amino acids of a target protein. Protein kinases regulate various aspects of cellular functions including cell survival, apoptosis, cell division, and metabolism via reversible and tightly regulated phosphorylations (*Figure 1A*; *Bhullar et al., 2018*, *Blume-Jensen and Hunter, 2001*, *Manning et al., 2002*). In addition to kinase phosphotransferase activities, kinase domains display scaffolding functions (*Shrestha et al., 2020*, *Reiterer et al., 2014*).

On the cellular level, kinases act as molecular switches, adopting conformational states that align with an active (ON) or inactive (OFF) kinase state (*Huse and Kuriyan, 2002*, *Yamaguchi and Hendrickson, 1996*, *Lopez et al., 2020*, *Feichtner et al., 2022*). These ON and OFF states of protein kinases signify a switch-like behavior which is governed by a collection of molecular mechanisms (*Figure 1A*). Kinases integrate, amplify and relay a wide variety of input signals, underscoring their pivotal role in numerous signaling pathways that often involve interconnected kinase regulation. Thus, one kinase directly modulates the activity level and role of the subsequent kinase in a cascade-like structure (*Guo et al., 2020*, *Avruch et al., 2001*). This regulation is facilitated by the formation of multi-protein complexes to spatiotemporally control and amplify signal transmission (*Morrison, 2001*, *Pouysségur et al., 2002*). *Figure 1A* highlights several factors affecting kinase functions. Selected modes of kinase regulation which are of relevance for the presented study are listed below.

First, the activity of a kinase can be altered by post-translational modifications (PTMs) of selected amino acids. PTMs alter kinase characteristics like its activity status, its location within the cell, rate of degradation, and its associations with other proteins (*Chou, 2020*, *Cohen, 2000*, *Deribe et al., 2010*).

Second, the scaffolding functions of kinase domains accompany the process of phosphotransferase reactions. These play central roles in the activation and deactivation process of interacting kinase protomers. This feature is key for pseudokinase activities, for relaying signaling inputs without catalytic functions (*Weinlich and Green, 2014*, *Morrison and Davis, 2003*, *Boudeau et al., 2006*).

Third, kinase functions depend on decisive regulatory protein interactions or movements for which several modes of regulation have been described. Intramolecular auto-inhibitory modules (AIMs) alter kinase activity states by reducing the accessibility of the substrate protein for the subsequent kiss-and-run phosphotransferase reaction (*Mayrhofer et al., 2020*, *Pufall and Graves, 2002*, *Xu et al., 2002*). In some cases, kinase domain activities are controlled through inter- or intramolecular interactions with substrate-like sequences. Pseudo-substrate stretches bind to the catalytic cleft of the kinase and hinder the phosphorylation of the substrate. This binding to the catalytic site is altered in response to input signals and manifested in kinase conformational changes, which in turn coordinate protein kinase activation cycles (*Schmitt et al., 2022*, *Kemp et al., 1994*). Besides intramolecular inhibition both activating and inactivating regulatory protein interactions exist and have been described for prototypical kinases such as PKA and CDKs (*Boudeau et al., 2003*, *Taylor et al., 2005*).

Fourth, kinase regulation is strongly dependent on the expression pattern and how the protein is stabilized in regard to the cell fate within the respective cell system or compartment (*Capra et al., 2006*).

Fifth, regulatory protein interactions of kinases depend on small molecule interactions. Besides different types of second messengers (e.g. $Ca^{2+}$, IP3, cAMP) (*Newton et al., 2016*, *Kasai and Petersen, 1994*) a collection of metabolites and ions contribute in orchestrating cellular kinase functions (*Ramms et al., 2021*).

Alterations of the ON and OFF states of kinases are also pertinent to kinase-related disorders. Deregulation of kinase functions is associated with the development of numerous diseases, such as cancer, inflammatory, infectious, and degenerative conditions (*Shchemelinin et al., 2006*, *Köstler and Zielinski, 2015*, *Ferguson and Gray, 2018*). Kinase malfunctions result from gene mutations, deletions, fusions, or increased or aberrant expressions. These mechanisms result in gain or loss of function of the involved kinase pathways, thus driving disease etiology and progression (*Ochoa et al., 2018*, *Van et al., 2021*, *Cicenas et al., 2018*).

For this technical report, we have incorporated disease-relevant full-length phosphotransferases which exhibit different modes of regulation into the kinase conformation (KinCon) reporter system. The KinCon reporter platform is a *Renilla* luciferase (*R*Luc) based protein-fragment complementation assay, which is based on the fusion of two fragments of *R*Luc to the N and the C terminus of a full-length kinase. This reporter can be used to track dynamic changes of kinases conformations due to mutations, PTMs, protein:protein interactions (PPIs), or binding of bioactive small molecules. We discuss the KinCon reporter principle at the beginning of the Results section. The kinases listed below

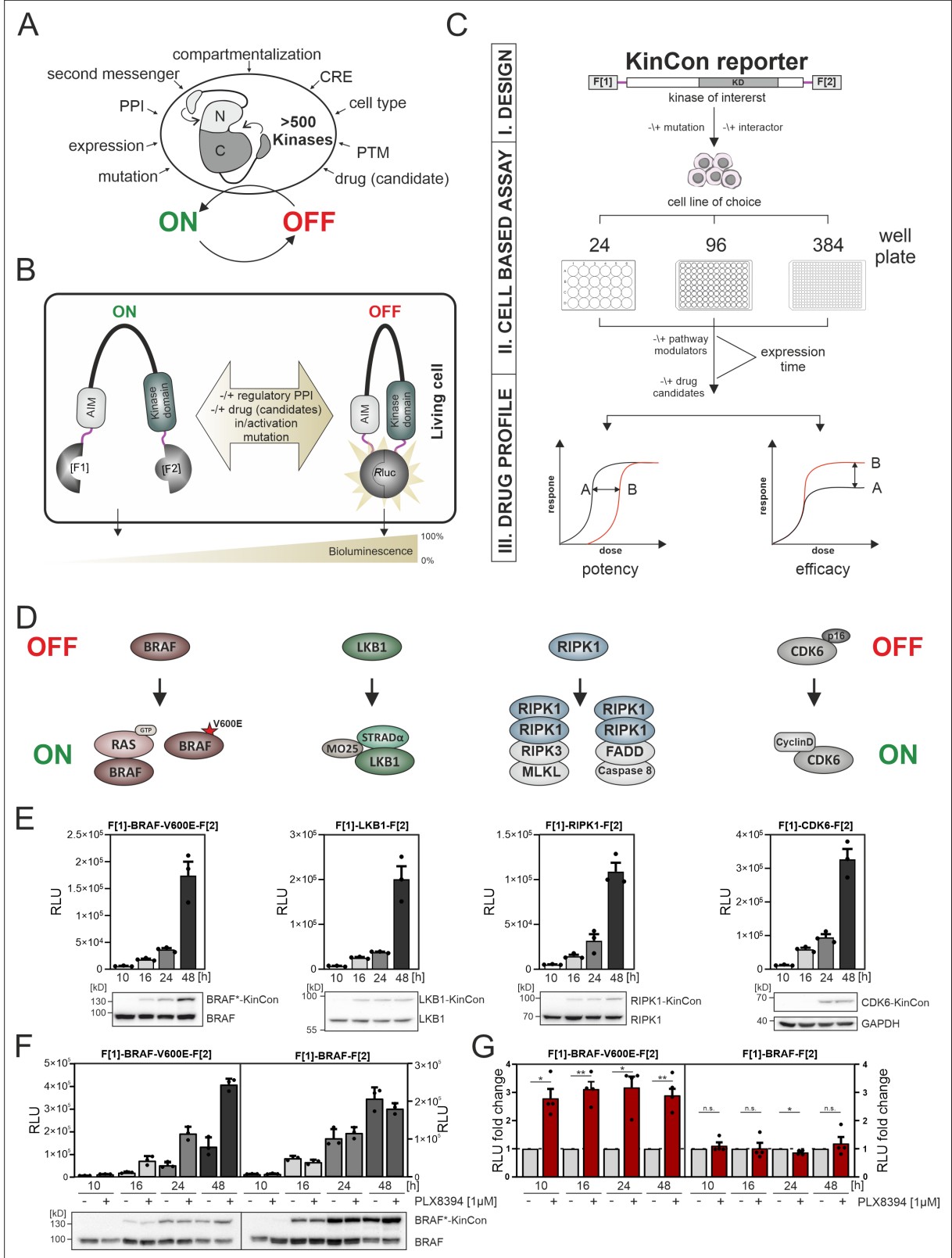

**Figure 1.** Kinase regulation and kinase conformation (KinCon) reporter technology features. (**A**) Impact of indicated factors/features (e.g. protein-protein interactions [PPIs], post-translational modifications [PTM], cis-regulatory elements [CRE]) on the switch-like behavior of kinases. (**B**) Schematic representation of the KinCon reporter technology using the *Renilla* luciferase (*R*Luc) protein-fragment complementation assay (PCA) as it applies to kinases such as BRAF which contain auto-inhibitory modules (AIM); *R*Luc fragments 1 and 2 are N and C terminally fused to the kinase of interest (with

*Figure 1 continued on next page*

*Figure 1 continued*

interjacent linker in red) and are labeled with F[1] and F[2]. PPIs, drug (candidate) or small molecule binding, mutations and/or PTMs may convert the KinCon reporter into different conformation states. Protein movements are quantified through measuring alterations of bioluminescence signals upon *R*Luc substrate addition. (**C**) Shown is the workflow for the KinCon reporter construct engineering and analyses using KinCon technology. The kinase gene of interest is inserted into the multiple cloning site of a mammalian expression vector which is flanked by respective PCA fragments (F[1]-, -F[2]; KD, kinase domain) and separated with interjacent flexible linkers. Expression of the genetically encoded reporter in indicated multi-well formats allows to vary expression levels and define a coherent drug treatment plan. Moreover, it is possible to alter the kinase sequence (mutations) or to co-express or knock down the respective endogenous kinase, interlinked kinases, or proteinogenic regulators of the respective pathway. After systematic administration of pathway modulating drugs or drug candidates, analyses of KinCon structure dynamics may reveal alterations in potency, efficacy, and potential synergistic effects of the tested bioactive small molecules (schematic dose-response curves are depicted). (**D**) Simplfied schematic representation of the activation mechanisms of BRAF, LKB1, RIPK1, and CDK6 complexes (with indication of selected regulators or complex components) engaged in altering OFF (top) or ON (bottom) kinase states. (**E**) Representative KinCon experiments of time-dependent expressions of indicated KinCon reporter constructs in HEK293T cells are shown (mean ± SEM). Indicated KinCon reporters were transiently over-expressed in 24-well format in HEK293T cells for 10 hr, 16 hr, 24 hr, and 48 hr each. Immunoblotting show expression levels of endogenous kinases and over-expressed KinCon reporters. (**F**) Impact of 1 µM PLX8394 exposure (for 1 hr) on BRAF and BRAF-V600E KinCon reporters (HEK293T cells) is shown. Representative experiment of n=4 independent is presented. (**G**) *R*Luc PCA values have been normalized on the untreated conditions. The mean ± SEM of PLX8394 exposure on BRAF conformation opening and closing of n=4 experiments is shown. RLU, relative light units. Statistical significance for G: one-sample t-test (*$p<0.05$, **$p<0.01$, ***$p<0.001$).

The online version of this article includes the following source data and figure supplement(s) for figure 1:

**Source data 1.** Raw unedited western blots shown in *Figure 1*.

**Source data 2.** Uncropped and labelled western blots shown in *Figure 1*.

**Figure supplement 1.** Time-dependent kinase conformation (KinCon) expression.

are characterized by distinctive modes of regulation, are viable targets for inhibition, or are difficult to assess directly through traditional biochemical measurements: the liver kinase B1 (LKB1, STK11), the receptor-interacting serine/threonine-protein kinase 1 (RIPK1), and the cyclin-dependent kinases 4 and 6 (CDK4/6).

LKB1 is active as trimeric cytoplasmic protein complex. It is the upstream regulator of the AMP-activated protein kinase (AMPK) (*Rodríguez et al., 2021*). LKB1 promotes AMPK signaling by forming an heterotrimeric complex with the pseudokinase STE20-related adaptor alpha (STRADα) and the scaffolding protein Mouse protein-25 (MO25) through phosphorylation and thus activation of cytoplasmic AMPK (*Boudeau et al., 2004*, *Narbonne et al., 2010*). Mutations in LKB1 can lead to the autosomal dominant disease Peutz-Jeghers syndrome (PJS) (*Mehenni et al., 1998*, *Beggs et al., 2010*). Inactivating mutations of LKB1 are frequently observed in non-small-cell lung cancer (NSCLC), cervical carcinoma, and malignant melanoma (*Wingo et al., 2009*, *Ndembe et al., 2022*).

RIPK1 acts as a central stress sensor to control cell survival, inflammation, and cell death signaling (*Clucas and Meier, 2023*). Deregulation of RIPK1- and RIPK3-involved signaling cascades have been linked to inflammatory bowel disease, rheumatoid arthritis, autoimmune conditions, and neuroinflammatory diseases such as Alzheimer's and Parkinson's disease (*Martens et al., 2020*, *Li et al., 2019*, *Speir et al., 2021*, *Clucas and Meier, 2023*). This is believed to be related to deregulations of both catalytic and kinase scaffolding functions. Thus, a collection of small molecule blockers has been identified to interfere with RIPK1 signaling and function. It's intriguing to note that RIPK1 and RIPK3 may share a similar mechanism of action with BRAF, as their structural arrangements and the dimerization of their kinase domains bear resemblance (*Raju et al., 2018*). In addition to this, auto-phosphorylation of RIPK1 and RIPK3 represents a central regulatory element, overall showing some similar molecular ties to BRAF (*Laurien et al., 2020*).

Unlike the auto-inhibitory mechanisms of kinase regulation, some kinases dynamically engage with activating and deactivating polypeptides (*Zhang et al., 2021*). One of the best studied examples is the interaction of regulatory proteins with the catalytic subunits of the cAMP-controlled serine/threonine protein kinase A (PKA) (*Taylor et al., 2012*, *Zhang et al., 2020*). In a similar manner CDKs form active and inactive protein complexes to directly promote the cell cycle (*Goel et al., 2018*). The central role of CDK4/6 lies in regulating cell cycle progression by phosphorylating and activating the key substrate retinoblastoma protein (Rb) that promotes G1 to S phase transition. On the molecular level this is controlled by complex formation of CDK4 and CDK6 with regulatory polypeptides (*James et al., 2008*). Binding to p16$^{INK4a}$, one of the most frequently mutated tumor suppressor proteins,

blocks CDK4/6 functions (*Quelle et al., 1997*). Cancer mutations in p16$^{INK4a}$ counteract this. As a result, kinase activating cyclinD proteins bind to CDK4/6 to promote carcinogenesis (*VanArsdale et al., 2015*). Thus, CDK4/6 inhibitors (CDK4/6i) found the way into the clinic, in particular for treating breast cancer patients (*Nebenfuehr et al., 2020*; *Yu et al., 2006*). The development of drug resistance upon CDK4/6 inhibitor therapies underscores the need for personalized treatments that consider the patient's genetic profile and the underlying alterations of CDK4/6 complexes present in their cancer cells (*Álvarez-Fernández and Malumbres, 2020*, *Knudsen and Witkiewicz, 2017*).

Various cellular mechanisms are employed to control the molecular switch-like behavior of protein kinases to temporally lock them into either an active or inactive conformation. These mechanisms include the PTMs of specific residues, the binding of regulatory proteins or co-factors, and allosteric changes induced by ligands (*Bhullar et al., 2018*, *Taylor et al., 2021*, *Newton, 2001*). This precisely controlled mode of action can be hindered, among others by patient-specific mutations or modified by bioactive small molecules (*Figure 1A*). Conventional methods often fall short in capturing the dynamics of kinase phosphotransferase and scaffolding activities which occur in their native cellular environments (*Klaeger et al., 2017*, *Croce et al., 2019*). This underscores the importance of advanced biotechnology-centered strategies for a comprehensive understanding of drug target engagement and protein binding in the intact cell. Here, we show that genetically encoded KinCon reporters are extendable to many more kinases, to enable systematic monitoring of cellular kinase activity states in living cells. In addition to its predictive capabilities, the KinCon technology serves as a valuable tool for uncovering cellular factors influencing drug efficacy (*Mayrhofer et al., 2020*, *Fleischmann et al., 2023*, *Röck et al., 2019*). Such insights into the molecular structure dynamics of kinases in intact cells upon interactions with small molecule inhibitors or regulatory proteins is necessary for the design of more effective therapeutic strategies.

## Results

Kinases act as molecular switches to integrate, amplify, restrict, and/or relay signal propagation in spatiotemporal fashion and in cell-type-specific manner. The precise coordination of usually oscillating kinase activities is a prerequisite for proper signal transmission (*Taylor et al., 2012*, *Pan and Heitman, 2002*). When investigating pathological kinase functions for therapeutic purposes, it is crucial to take the cellular elements that influence kinase activities into account. In *Figure 1A* we list a collection of factors which affect kinase activity states. It is thus essential to closely monitor the physiological and pathophysiological kinase functions of activation and deactivation in intact cell settings in the presence and absence of patient mutation and drug exposure.

One cell-based technology for studying cellular kinase activity states is the KinCon reporter technology (*Röck et al., 2019*, *Mayrhofer et al., 2020*, *Fleischmann et al., 2023*). It is a highly sensitive assay for tracking kinase activities which is mirrored by the alterations of full-length kinase structures in cells. With in silico predictions it was projected that more than 200 protein kinases of the human kinome contain cis-regulatory elements. In many cases these sequence stretches act as auto-inhibitory modules, so called AIMs (*Mayrhofer et al., 2020*, *Yeon et al., 2016*). In *Figure 1B* we depict exemplary how underlying kinase ON and OFF states can be tracked using KinCon reporter technology (*Enzler et al., 2020*). The readout is based on the molecular motion of the full-length kinase containing diverse cis-regulatory elements / AIMs. Cell-type-specific KinCon measurement allows the testing of the influence of mutations, signaling pathway activation, binary protein interactions, and drug binding on the respective reporter reflecting conformational changes of full-length kinases. We have previously applied the technology to gain insights into the functioning of two kinases that belong to the MAPK pathway. In these proof-of-concept studies, we showed that BRAF and MEK1 KinCon reporters are direct real-time readouts for kinase activities in intact cell settings caused or altered by mutation and drug treatments (*Röck et al., 2019*, *Mayrhofer et al., 2020*, *Fleischmann et al., 2021*, *Fleischmann et al., 2023*).

The construction principle of the KinCon reporter is modular. For the generation of the genetically encoded KinCon reporter the sequence of the kinase of choice is inserted into the multiple cloning site (MCS) of a mammalian expression construct. The MCS is flanked by the coding regions of a split luminescent protein for cellular over-expression experiments of the encoded hybrid reporter protein (see the KinCon reporter protein domain structure at the top of *Figure 1C*). In many cases it is

sufficient to fuse to the N and C terminus of the full-length kinase sequence the two fragments of the respective reporter protein (with intervening flexible linker stretches shown in red, *Figure 1B and C*).

In this study we used the protein-fragment complementation fragments (PCA, F[1]- and -F[2]) of the *Renilla* luciferase (*R*Luc-PCA) (*Stefan et al., 2007*). The KinCon reporters are constructed to facilitate the intramolecular complementation of appended RLuc PCA fragments. Transient expression offers the flexibility to analyze different time frames for KinCon reporter expression and drug candidate exposures, in either low- or high-throughput format in intact cells (*Figure 1C*). Besides applying wild-type (wt) reporters it allows for the 'personalization' of the sensor setup by integrating patient-specific mutations, co-expressing regulatory proteins, or making systematic changes to PTM sites.

Following KinCon reporter expression along with co-expression of interacting molecules in the appropriate cell plate format, systematic perturbations can be applied. Following addition of the lucif-erase substrate to cells grown in a mono-layer or in suspension, cellular bioluminescence signals are emanating from complemented *R*Luc PCA fragments (*Figure 1B and C*; *Röck et al., 2019*, *Mayrhofer et al., 2020*). Light recordings and subsequent calculations of time-dependent dosage variations of bioluminescence signatures of parallel implemented KinCon reporter configurations aid in estab-lishing dose-response curves. These curves are used for discerning pharmacological characteristics such as drug potency, effectiveness of drug candidates, and potential drug synergies (*Figure 1C*). In order to enhance our understanding of kinase structure dynamics we selected a group of kinases which activities are altered in different pathological settings. These examples emphasize how muta-tions, PTMs, PPIs, or kinase drugs induce context-dependent effects on the conformation states of kinases. In *Figure 1D* we present a schematic and simplified depiction of the kinase's ON and OFF conditions for complexes emanating from the kinases BRAF, LKB1, RIPK1, and CDK6. Exemplary, we show how protein complex formation and patient mutations contribute or perturb kinase activation cycles. As a starting point, we illustrated the high sensitivity of the reporter system for tracking basal activity conformations of the kinases BRAF-V600E, LKB1, RIPK1, and CDK6 respectively. We showed that transient over-expression of these KinCon reporters for a time frame of 10 hr, 16 hr, 24 hr, or 48 hr in HEK293T cells delivers consistently increasing signals for all KinCon reporters (*Figure 1E*, *Figure 1—figure supplement 1A*). Immunoblotting of cell lysates following luminescence measure-ments showed expression levels of the reporters in the range and below the endogenous expressed kinases (*Figure 1E*).

Next, we analyzed the BRAF kinase activity conformations using wild type (wt) and mutated KinCon BRAF reporters. The V600E mutation, found primarily in melanoma patients, effectively immobilized the respective BRAF KinCon reporter in its opened and active conformation (*Davies et al., 2002*, *Lavoie et al., 2020*, *Karoulia et al., 2017*, *Lito et al., 2013*, *Röck et al., 2019*). Previously we have shown that FDA-approved melanoma drugs (Vemurafenib, Encorafenib, Dabrafenib) and one drug candidate from clinical studies (PLX8394) converted the opened BRAF-V600E reporter back to the more closed and thus inactive conformation (*Röck et al., 2019*, *Mayrhofer et al., 2020*, *Yao et al., 2019*). Using this readout, we showed that at expression levels of the BRAF KinCon reporter below the immunoblotting detection limit, 1 hr of drug exposure exclusively converted BRAF-V600E to the more closed conformation (*Figure 1F and G*, *Figure 1—figure supplement 1B*). These data under-line that at expression levels far below the endogenous kinase, protein activity conformations can be tracked in intact cells. This may represent the more authentic (patho)physiological context and takes the molecular interactions with endogenous factors into consideration.

Next, we adapted the KinCon biosensor technology to investigate the correlation between confor-mation and activity regulation of key kinase pathways. We analyzed three different kinase pathways displaying different modes of kinase ON-OFF regulation.

## Trimeric LKB1 complexes

Upon heterotrimeric complex formation with the pseudokinase STRADα and the scaffolding protein MO25, the kinase LKB1 contributes to the activation of AMPK by phosphorylation at the position Thr172 (*Figure 2A*; *Shackelford and Shaw, 2009*, *Boudeau et al., 2004*, *Baas et al., 2003*). The pseudokinase domain of STRADα directly interacts with the kinase domain of LKB1, thus triggering the activation of LKB1's tumor-suppressing phosphotransferase functions (*Figure 2B and C*). It is assumed that upon ATP-binding STRADα occupies an active conformation. In this scenario LKB1 affin-ities for binding rise and it binds as a pseudo-substrate. MO25 acts as scaffold for the kinase dimer to

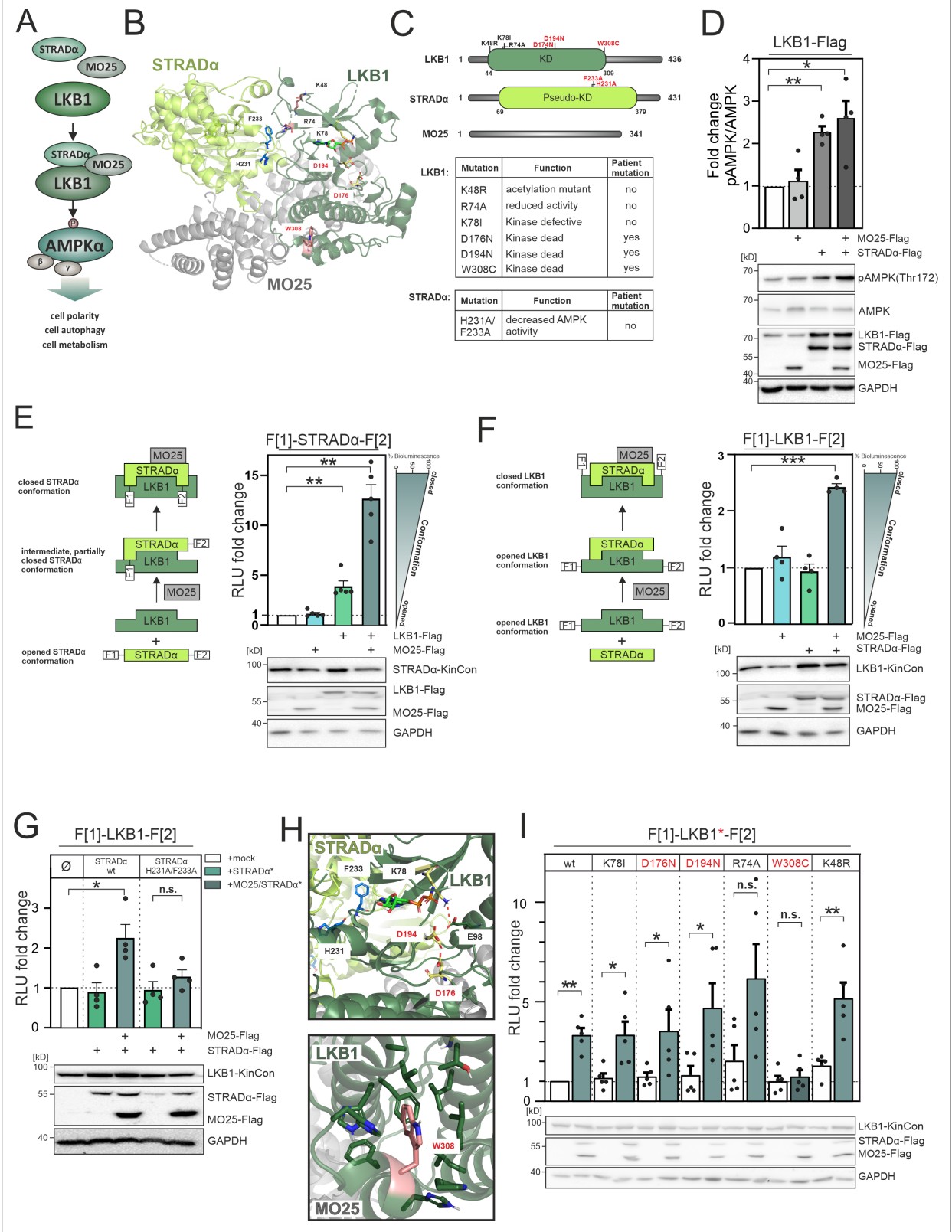

**Figure 2.** LKB1 emanating complexes and mutation-related kinase activity conformations in intact cells. (**A**) Simplified representation of the LKB1-complex composition which promotes AMPKα signaling via phosphorylation at position Thr172. (**B**) Crystal structure of the LKB1-STRADα-MO25 complex (PDB code 2WTK, *Zeqiraj et al., 2009a*) representing a snapshot of trimeric complex assembly. The missense mutations we have analyzed are indicated in blue (STRADα) and pale yellow and rose (LKB1). The ATP analogue AMP-PNP is depicted in light green sticks. (**C**) Domain organization of

*Figure 2 continued on next page*

*Figure 2 continued*

human LKB1, STRADα, and MO25 (accession numbers: Q15831, Q7RTN6, Q9Y376) with indication of the kinase and pseudokinase domains (KD). Shown in red are tested missense mutations. These are summarized in the table together with their origin and assumed functions (*Zubiete-Franco et al., 2019*, *Qing et al., 2022*, *Yang et al., 2019*, *Ui et al., 2014*, *Al Bakir et al., 2023*, *Islam et al., 2019*, *Boudeau et al., 2004*). (D) Effect of co-expressions of indicated kinase complex components on AMPK phosphorylation (HeLa cells, 48 hr post transfection) (mean ± SEM, n=4 ind. experiments; 3x-Flag is indicated as flag). (E) Illustration of the kinase conformation (KinCon) reporter setup for STRADα KinCon measurements: Effect of LKB1-STRADα-MO25 complex formation on the STRADα KinCon reporter opening and closing (HEK293T cells, 48 hr post transfection). Expression corrected signals (STRADα-KinCon) are shown (mean ± SEM, n=4 ind. experiments). (F) KinCon reporter setup for LKB1 KinCon measurements: Effect of LKB1-STRADα-MO25 complex formation on the LKB1 KinCon reporter conformation. Expression corrected signals are shown (LKB1-KinCon; HEK293T cells, 48 hr post transfection) (mean ± SEM, n=5 ind. experiments). (G) LKB1-KinCon measurements upon co-expression of indicated proteins displaying the binding deficient STRADα mutations H231A/F233A (HF; see binding interface in B and H). Expression corrected signals are displayed (HEK293T cells, 48 hr post transfection) (mean ± SEM, n=4 ind. experiments). (H) Structure depiction highlights the localization of mutations conferring altered LKB1 functions. LKB1 residues K78, D176, and D194 (pale yellow sticks) are located within the catalytic cleft and in close proximity to AMP-PNP (light green sticks). (I) Impact of LKB1 missense mutations (three patient mutations D176N, D194N, and W308C and three 'non-patient' mutations K48R, R74A, K78I) on KinCon conformational changes upon co-expression of interactors. Expression corrected signals are displayed (HEK293T cells, 48 hr post transfection) (mean ± SEM, n=4 ind. experiments). Statistical significance for D, E, F, G, and I: one-sample t-test (*p<0.05, **p<0.01, ***p<0.001).

The online version of this article includes the following source data and figure supplement(s) for figure 2:

**Source data 1.** Raw unedited western blots shown in *Figure 2*.

**Source data 2.** Uncropped and labelled western blots shown in *Figure 2*.

**Figure supplement 1.** Complex formation of LKB1/STRADα/MO25 in HeLa and SW480 cells.

**Figure supplement 1—source data 1.** Raw unedited western blots shown in *Figure 2—figure supplement 1*.

**Figure supplement 1—source data 2.** Uncropped and labelled western blots shown in *Figure 2—figure supplement 1*.

promote the activated LKB1 conformation state (*Zeqiraj et al., 2009a*, *Zeqiraj et al., 2009b*). Further, MO25 binding stabilizes this trimeric cytoplasmic complex (*Figure 2A and B*; *Boudeau et al., 2003*, *Baas et al., 2003*).

The capacity of STRADα for allosterically modulating LKB1 functions through direct interaction underlines that pseudokinases are more than inert scaffolds. They are involved in modulating interacting enzyme entities (*Rajakulendran and Sicheri, 2010*, *Reiterer et al., 2014*). This is in line with the common belief that pseudokinases can employ switch-like transitions to regulate signaling networks (*Kung and Jura, 2019*, *Shrestha et al., 2020*). Given the discovery of LKB1 inactivating mutations in diseases like PJS, NSCLC, and colorectal cancer, there is a growing interest in exploring strategies to therapeutically restore the function of mutated LKB1 (*Sanchez-Cespedes, 2007*, *Launonen, 2005*, *Kitajima et al., 2019*).

In contrast to blockers of kinase functions it is more challenging to identify activator molecules for promoting reactivation of the LKB1-AMPK axis. Thus, we investigated the impact of trimeric complex formation of LKB1:STRADα:MO25 on downstream activity of LKB1. We adopted the KinCon technology for measuring involved kinase conformation states. First, we transiently over-expressed the three flag-tagged polypeptides LKB1, STRADα, and MO25 in HeLa cells. Next, we employed the pT172-AMPK/AMPK ratio as a measure of cellular LKB1 activity, observing the highest increase of LKB1-mediated downstream phosphorylation of AMPKα-T172 following co-expression of both, MO25 and STRADα, respectively. Using the chosen cellular setting and transfection protocol with a 1:1:1 ratio of transfected expression constructs, a similar effect was observed when solely over-expressing STRADα. No impact on AMPK phosphorylation was observed when MO25 was coexpressed alone (*Figure 2D*).

Next, we started to investigate the impact of LKB1 and MO25 co-expression on STRADα conformation changes. Notable is the fact that only faint STRADα KinCon signals were detected in the absence of MO25 and LKB1 co-expression. To our surprise we have observed a more than 10-fold elevation of the reporter signals following co-expression of both interacting partners in HEK293T cells (48 hr of expression, *Figure 2E*, *Figure 2—figure supplement 1A*). This data supports the notion that STRADα engages an opened conformation, where C and N termini are separated and thus almost no bioluminescence signals can be detected under basal conditions. Upon interaction with LKB1 this conformation shifts to a partially closed intermediate state. The trimeric complex further promotes structure closing. We have observed the same tendency – but to a lower extent – using LKB1 KinCon

readouts in the presence and absence of both interacting proteins (*Figure 2F*, *Figure 2—figure supplement 1B*).

To validate these findings, we then tested the impact of the LKB1-binding deficient STRADα-H231A/ F233A (HF) mutant proteins (*Boudeau et al., 2004*) in co-expression experiments with the LKB1 KinCon reporter. It has previously been reported that these mutations prevent STRADα-LKB1 dimer and greatly hinders STRADα-MO25-LKB1 trimer formation and thereby AMPK activation (*Zeqiraj et al., 2009a*). We observed that the so-called HF double mutation of STRADα is sufficient to abolish the elevating effect of trimer complex formation (*Figure 2G*). Indeed both mutations are located in the LKB1/STRADα binding interface, thereby preventing complex formation (*Qing et al., 2022*, *Zeqiraj et al., 2009a*; *Figure 2H*). These findings underline that trimeric complex formation alters the opening and closing of the tested full-length kinase structures using the applied KinCon reporter readout. With this reporter technology we monitored the interaction controlled activities of LKB1 directly in a living cell setting.

LKB1 loss-of-function mutations have been identified in a plethora of pathological conditions including autosomal diseases and many different forms of cancers (*Sanchez-Cespedes, 2007*, *Launonen, 2005*, *Molaei et al., 2022*). Thus, we set out to analyze the impact of patient mutations on alterations of LKB1 kinase conformations.

In *Figure 2C* we have listed the proposed functions of the tested mutations. Three 'tool' mutations (K48R, R74A, and K78I) (*Zubiete-Franco et al., 2019*, *Qing et al., 2022*, *Yang et al., 2019*) and three patient mutations (D176N, D194N, and W308C, in red) (*Ui et al., 2014*, *Al Bakir et al., 2023*, *Islam et al., 2019*) were analyzed.

*Figure 2H* highlights the location of these mutations within LKB1. K78, D176, and D194 are highly conserved residues within the ATP-binding pocket and are critical for kinase activity: K78 forms a salt bridge with E98 on the αC-helix which stabilizes a functional active conformation, the catalytic D176 is crucial for phosphoryl transfer, and D194 is part of the DFG motif involved in binding of the $Mg^{2+}$ ion (*Fabbro et al., 2015*, *Meharena et al., 2016*). The residue W308 is part of a hydrophobic cluster and thus surrounded by lipophilic residues (*Zeqiraj et al., 2009a*). R74 is located within the STRADα binding interface, and has been reported to interact with STRADα-Q251 (*Zeqiraj et al., 2009a*), and K48 is a solvent-exposed acetylation site (*Zubiete-Franco et al., 2019*, *Zeqiraj et al., 2009a*) located on the back of the N-lobe. Among these mutations, only the W308C mutation prevented significant closing of the LKB1 conformation when co-expressed with STRADα and MO25 (*Figure 2I*). This indicates that the catalytic site residues are critical for enzymatic activity, but play a less important role in maintaining the active LKB1 conformation in the presence of both STRADα and MO25. The W308C mutation has previously been identified to cause a reduction in LKB1 stability when compared to the wt variant (*Islam et al., 2019*), suggesting the polar side chain of cysteine likely disrupts the interaction network within the hydrophobic cluster. Additionally, the LKB1 mutant W308C diminishes its catalytic activity (*Islam et al., 2019*, *Boudeau et al., 2004*, *Mehenni et al., 1998*), consistent with the observed prevention of the PPI-directed closing of the LKB1 KinCon reporter conformation (*Figure 2I*). These findings suggest that LKB1-W308C lost its ability to form the heterotrimeric complex, with implications on hindering downstream activation and thus affecting its tumor suppressor functions. Overall these findings underline that the KinCon technology can be extended to track the impact of binary protein complexes and related cancer mutations on kinase activity dynamics. Moreover, this data demonstrated that in contrast to the previously published MEK1 and BRAF KinCon reporter (*Fleischmann et al., 2021*, *Fleischmann et al., 2023*, *Röck et al., 2019*, *Mayrhofer et al., 2020*), the more closed STRADα and LKB1 KinCon reporter conformations represent the more active full-length kinase states.

## Mutations and inhibitors induce RIPK1 conformational changes

RIPK1 is a multi-functional protein and central regulator of cell death, inflammatory processes, and immune responses (*Figure 3A*; *Degterev et al., 2019*, *Clucas and Meier, 2023*, *Silke and Meier, 2013*). Genetic alterations of RIPK1 are linked to immune and autoinflammatory diseases (*Lalaoui et al., 2020*, *Tao et al., 2020*). Small molecule-mediated inhibition of RIPK1 kinase activity was shown to counteract the necroptotic phenotype in disease models (*Tao et al., 2020*). This exemplifies the potential of RIPK1 targeting drugs for the treatment of certain inflammatory diseases.

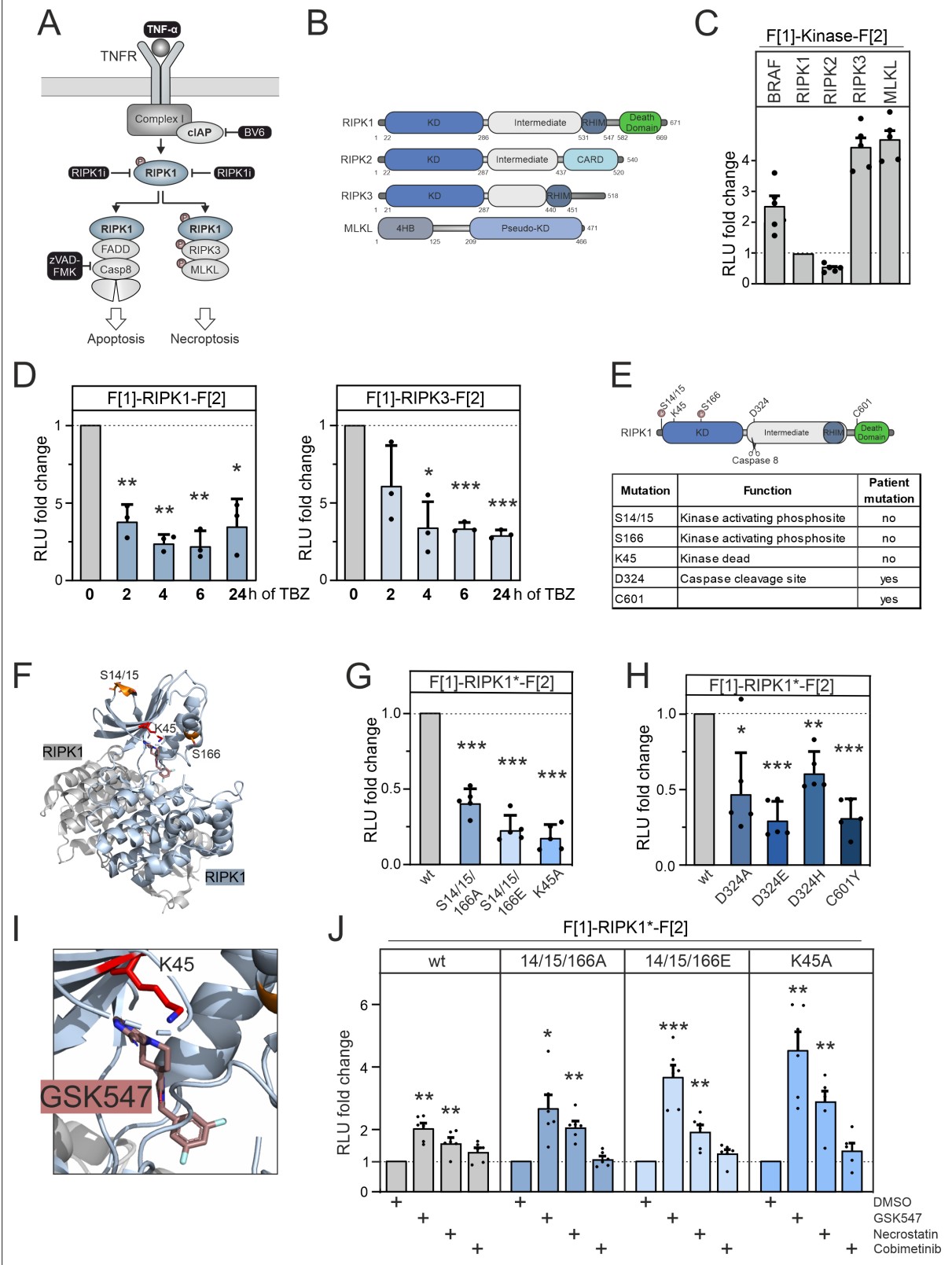

**Figure 3.** RIPK1 conformation dynamics. (**A**) Simplified schematic representations of the activation pathways for apoptosis and necroptosis. Highlighted in black is the combination treatment termed TBZ (10 pg/ml TNFα, 10 nM BV-6, and 20 nM zVAD-FMK) that induces necroptosis. (**B**) Domain organization of human RIPK1 (accession number: Q13546), RIPK2 (accession number: O43353), RIPK3 (accession number: Q9Y572), and MLKL (accession number: Q8NB16). (**C**) Basal signals of indicated kinase conformation (KinCon) reporters following transient over-expression in HEK293T cells. Bars

*Figure 3 continued on next page*

*Figure 3 continued*

represent the RLU fold change relative to RIPK1 (mean ± SD, n=5 ind. experiments). (**D**) Time-dependent treatments using TBZ of HEK293T cells transiently expressing wild-type (wt) RIPK1 (left) and wt RIPK3 (right) KinCon reporters (expression corrected) (mean ± SD, n=3 ind. experiments). (**E**) Domain organization of RIPK1 displaying missense mutation sites. (**F**) 3D structure of RIPK1 dimers with functional mutations highlighted (PDB code: 6HHO, *Wang et al., 2018*). GSK547 is depicted as brown sticks. (**G**) KinCon reporter signals with/without mutations (S14/15/166A, S14/15/166E, K45A) were measured in a HEK293T RIPK1 knock-out (KO) cell line (expression corrected) (mean ± SEM, n=5 ind. experiments). (**H**) KinCon reporter signals of RIPK1 (patient loci: D324A, D324E, D324H, C601Y) were measured in HEK293T RIPK1 KO cells (expression corrected) (mean ± SD, n=5 ind. experiments). (**I**) 3D structure of RIPK1 with the inhibitor GSK547, which binds to an allosteric site in close proximity to the ATP-binding site (PDB code: 6HHO, *Wang et al., 2018*). (**J**) RIPK1 reporter signals with indicated mutations (described in G) upon exposure to GSK547 and Necrostatin 1 µM, and the MEKi Cobimetinib (1 µM, control experiment) or DMSO for 1 hr (mean ± SD, n=6 ind. experiments, HEK293T RIPK1 KO). Statistical significance for C–J: one-sample t-test (*p<0.05, **p<0.01, ***p<0.001).

The online version of this article includes the following source data and figure supplement(s) for figure 3:

**Source data 1.** Raw unedited western blots underlying *Figure 3* and shown in *Figure 3—figure supplement 1*.

**Source data 2.** Uncropped and labelled western blots *Figure 3* and shown in *Figure 3—figure supplement 1*.

**Figure supplement 1.** Small molecule effects on RIPK1.

**Figure supplement 2.** Conformation and phosphorylation of RIPK1.

**Figure supplement 2—source data 1.** Raw unedited western blots shown in *Figure 3—figure supplement 2*.

**Figure supplement 2—source data 2.** Uncropped and labelled western blots shown in *Figure 3—figure supplement 2*.

We established a KinCon reporter platform for monitoring the conformational changes of central kinases and pseudokinases involved in the necroptosis signaling pathway. Additionally, we examined the effects of binding of allosteric kinase blockers to both wt and mutant RIPK1 reporters. We depict the domain organization of RIPK1, RIPK3, and MLKL. We also included analyses of RIPK2, which is not part of the necroptosis pathway (*Figure 3B*). We cloned the human versions of these kinase genes into KinCon reporter expression constructs. After transient over-expression of these kinases in HEK293T cells, we quantified the basal *R*Luc PCA signals and compared these to the basal signal of the BRAF KinCon reporter (*Figure 3C*).

As starting point, we evaluated whether RIP kinases 1 and 3 activation would display conformational changes. We promoted necroptosis introducing stimuli by activating the TNFα pathway with TNFα (*Christofferson et al., 2012*), while preventing repressive RIPK1 ubiquitylation with the SMAC mimetic BV-6 (*Li et al., 2011*). To prevent the onset of apoptosis and instead induce activation of RIPK3, we blocked caspase 8 activation with the caspase inhibitor zVAD-FMK (*Festjens et al., 2007*) (as illustrated in *Figure 3A*).

We used HEK293T cells and transiently expressed the RIPK1 and RIPK3 KinCon reporters for 48 hr. Bioluminescence readouts of the KinCon reporters were quantified after indicated timings of TNFα, BV-6, and zVAD-FMK (TBZ) treatments. We observed successive accumulation of the respective KinCon reporter protein levels over the treatment time course (*Figure 3—figure supplement 1A*). We assume that this is caused by blocking of caspase 8, thus preventing the cleavage of both RIPK KinCon biosensors. As consequence we normalized the obtained *R*Luc PCA signals on reporter expression levels. We showed that both KinCon reporters exhibit markedly reduced bioluminescence in the necroptotic cellular environment (*Figure 3D*). These results support the notion that both kinases shift to a more opened ON state of the full-length protein upon pathway activation.

Conformational modulation of the RIPK1 KinCon following TNFα pathway activation suggests that the reporter is incorporated into complex I. We were interested to test the degree to which the RIPK1 KinCon represents a functional RIPK1 entity. For this purpose, we conducted co-expression experiments of various RIPK1 KinCon and flag-tagged RIPK1 variants in RIPK1-deficient HEK293T cells. We used auto-phosphorylation at S166 as the primary readout for RIPK1 catalytic activity. Our results showed that the RIPK1 KinCon reporter was not capable of auto-phosphorylation. In the presence of active flag-tagged RIPK1 however, trans-phosphorylation of the KinCon reporter hybrid proteins was evident (*Figure 3—figure supplement 2B*).

To further deepen our understanding of the modes of RIPK1 activation we integrated a collection of missense mutations into the KinCon reporter. Their assumed functions have been listed in *Figure 3E*. At the kinase N terminus a stretch of phosphorylation sites are attributed to kinase activation. Auto-phosphorylations at the positions S14, S15, and S166 have been shown to trigger

downstream activation (*Laurien et al., 2020*, *Wang et al., 2018*). The localizations are highlighted in the domain organization and the structure depiction of RIPK1 dimers (*Figure 3E and F*). It needs to be noted that additional kinases have been linked to control the phosphorylation and activity state of RIPK1 (*Dondelinger et al., 2019*, *Xu et al., 2018*, *Jaco et al., 2017*).

Auto-phosphorylation events have been described for the positions S14/S15/S166 (referred to as S14/15/166) and are critical for RIPK1 activation (*Wang et al., 2021*). Thus, we integrated mutations that mimic protein phosphorylation (S to E amino acid substitutions) and those that are deficient of phosphorylation (S to A amino acid substitutions), aiming to track their impact on molecular motions of RIPK1 to simulate the more activate kinase status. In addition, the kinase-deficient mutant K45A was investigated (*Shutinoski et al., 2016*). Interestingly, K45 in RIPK1 corresponds to K78 in LKB1, highlighting the importance of this lysine residue in the functionality of various kinases. Our KinCon findings unexpectedly demonstrated for RIPK1 that in addition to the kinase-deficient mutant (K45A) all the phosphorylation mimetic (S14/15/166E) and phosphorylation preventing (S14/15/166A) mutations tested led to an opening of RIPK1 conformations (*Figure 3G*).

We also incorporated prevalent patient mutations of RIPK1, observed in distinct inflammatory diseases, into the KinCon biosensor. Besides the patient mutations of the regulatory caspase 8 cleavage site in RIPK1 (D324A/E/H) we integrated the C601Y missense mutation, which causes immunodeficiency and inflammatory bowel disease, into the reporter (*Li et al., 2019*). Again, we observed an opening of RIPK1 conformations with all mutations tested (*Figure 3H*). This data supports the notion that in addition to preventing caspase 8 cleavage the mutations at position 324 affect the activity conformation of RIPK1. The C601Y substitution promotes RIPK1 opening as well.

We next tested all RIPK1 KinCon settings with the allosteric RIPK1 modulators GSK547 and Necrostatin (*Puylaert et al., 2022*, *Cho et al., 2011*). When bound to RIPK1, both of these compounds occupy a pocket located behind the ATP-binding site (*Figure 3I*, *Figure 3—figure supplement 1B*). The kinase-deficient mutation (K45A) of the KinCon biosensor displayed the strongest opening of the enzyme conformation. We used the RIPK1-K45A KinCon to benchmark the time-dependent effects of Necrostation and GSK547 on the conformational changes (*Figure 3—figure supplement 1B*). Application of both compounds showed engagement with all the tested RIPK1 KinCon reporters tested. An increase of bioluminescence for all RIPK1 inhibitors was evident reflecting the closing of the full-length kinase conformation. Even the phosphorylation mimetic (S14/15/166E) and phosphorylation preventing (S14/15/166A) KinCon reporters showed these phenomena of promoting a more closed configuration when compared to the unrelated small molecule control, the MEK1 kinase inhibitor Cobimetinib (*Figure 3J*, *Figure 3—figure supplement 1C*, *Figure 3—figure supplement 2A*). This data underlines that application of both allosteric kinase blockers alters the RIPK1 conformations and we assume scaffolding functions of active and inactive RIPK1 protomers.

## CDK4 and CDK6 interactions and conformations

CDK4 and CDK6 exhibit proto-oncogenic properties that rely on the presence and interaction with regulatory proteins (*Scheiblecker et al., 2020*, *Eferl and Wagner, 2003*). Further, CDK6 functions as a chromatin-bound factor and transcriptional regulator, thereby promoting the initiation of tumorigenesis (*Kollmann et al., 2013*, *Semczuk and Jakowicki, 2004*, *Scheicher et al., 2015*; *Kollmann and Sexl, 2013*). These activities are modulated by its molecular interaction partners, particularly cyclins and proteinogenic inhibitors (*Goel et al., 2018*, *Giacinti and Giordano, 2006*, *Ortega et al., 2002*). CDK4/6 deregulation is caused by alterations of protein expression levels and/or by mutations within regulatory proteins. Prime example is the tumor suppressor protein p16$^{INK4a}$ which is one of the most frequently mutated genes in human cancers (*Romagosa et al., 2011*, *Liggett and Sidransky, 1998*).

The CDK4/6-cyclinD protein complex has been found to be hyperactivated in many cancers and therefore promote tumor growth (*Choi et al., 2012*). This implies that CDK4/6 serves as a significant therapeutic target, and chemical inhibitors have been developed to target CDK4/6 phosphotransferase activities. Efficient CDK4/6 inhibitors such as Palbocilicb, Ribociclib, and Abemaciclib are currently in clinical use against breast cancer (*Otto and Sicinski, 2017*). A major drawback of these therapies is the emergence of drug resistance (*Fassl et al., 2022*). The development of precision medicine oriented polypharmacology therapies is one strategy to avoid the early advent of resistance (*Álvarez-Fernández and Malumbres, 2020*, *Yang et al., 2017*).

On the molecular level CDK4/6 activity states depend on the formation of PPIs with cyclinD or p16[INK4a], wherein cyclinD activates them, while p16[INK4a] inhibits kinase activity cycles of promoting Rb phosphorylation (*Figure 4A*). Distinct to the other kinases tested here, the CDK protein coding sequences are composed almost exclusively of the kinase domain with short additional stretches at the N and C termini (*Figure 4B*). In contrast to BRAF and RIPK1, the regulatory protein motifs are encoded by separate polypeptides. In the following, we tested the binary interaction of p16[INK4a] with CDK4 and CDK6. Structures of these complexes have been described. It is evident that substitution of arginine at the position 31 of CDK6 with cysteine (R31C) alters the binding affinities for p16[INK4a]. It forms two ionic interactions with p16[INK4a] residues D74 and D84, which are lost in the R31C mutant (*Figure 4C*). Experimental data further supports this notion (*Rodríguez-Díez et al., 2014*). We included the respective mutations by generating CDK6-R31C and also the corresponding R24C mutated expression construct of CDK4 (CDK4-R24C). We conceived a work program with PPI and KinCon reporters to elucidate any relations of dimerization and drug binding.

We started analyzing binary protein interaction of p16[INK4a] with CDK4 and CDK6. We fused the two *R*Luc fragments that are used in the KinCon reporter to two different proteins, in this case CDK4/CDK6 and p16[ink4a]. When two proteins interact, the two *R*Luc PCA fragments form a complemented luciferase and an increased light signal can be detected (*Figure 4D*, left). Originally the underlying technology had been developed for the dynamic measurements of regulatory kinase interactions of PKA (*Stefan et al., 2007*). Besides testing the kinase mutations CDK6-R31C and CDK4-R24C we generated the murine homolog of p16[INK4a] displaying the cancer mutation P40L. P40L showed reduced affinities for the tested CDKs and thus no longer inhibits phosphotransferase functions of CDK4/6 (*Yarbrough et al., 1999*). Upon complex formation of CDKs with p16[INK4a] the C terminal fused *R*Luc PCA reporter fragments complement and bioluminescence signals can be detected directly in intact cells transiently expressing the hybrid proteins. Using this cell-based PPI reporter we observed a significant reduction of p16[INK4a]:CDK4/6 complex formation when the respective R residues are mutated to C. Integration of the P40L mutation in p16[INK4a] prevented complex formation as expected and further validates the reporter system applied (*Figure 4D*, *Figure 4—figure supplement 1*).

Next, we set out to test the impact of these CDK mutations on kinase conformations. We have previously shown that basal CDK4/6 KinCon conformation states can be tracked using the KinCon reporter technology (*Mayrhofer et al., 2020*). Thus, we evaluated if a reduction of interaction with inhibitory proteins such as p16[INK4a] alters CDK full-length conformations. We transiently over-expressed the wt and mutant CDK4/6 KinCon reporter in HEK293T cells. After 48 hr of expression the cells were subjected to bioluminescence measurements. The results showed that the reduced affinity for p16[INK4a] binding alters the conformations of both CDKs. P16[INK4a] binding deficient CDK4 and CDK6 mutants exhibited a more opened conformation, as indicated by a decrease in bioluminescence (*Figure 4E*). In the following, all KinCon reporters have been subjected to analyses upon CDK4/6 inhibitor exposure (*Yam et al., 2018*). Again we used HEK293T cells and transiently expressed the wt and mutated KinCon reporters for 48 hr. We then exposed the cells to three clinically applied CDK4/6 inhibitors (CDK4/6i) at a concentration of 1 μM for 3 hr. Overall we could not detect any major effects of drug exposure on the dynamics of the four CDK KinCon reporters which showed different affinities for p16[INK4a] binding (*Figure 4F*). This data underlines that these breast cancer drugs which should bind to the active protein kinase did not induce full-length CDK4 and CDK6 conformational changes in the tested cell culture settings.

## Effect of different kinase inhibitor types on the KinCon reporter system

The inhibitors used in this study fall into different inhibitor categories (type I, type I 1/2, type II, and type III). The classification of inhibitors is determined by the activation state of the protein kinase, particularly the positioning of the DFG-Asp (active in, inactive out) and the αC-helix (active in, inactive out) (*Zhang et al., 2009*). However, type III (allosteric) inhibitors deviate from this pattern. They bind adjacent to the ATP-binding pocket, enabling simultaneous binding of ATP and the inhibitor (*Wu et al., 2015*). Type I inhibitors selectively bind to the active conformation of the kinase, characterized by DFG in and αC in. Type I 1/2 inhibitors, on the other hand, bind to the inactive state of the kinase, represented by DFG in and αC out. Type II inhibitors bind to the inactive kinase, where DFG is out and αC can be either in or out (*Arter et al., 2022*; *Figure 5A and B*). In this study we have observed

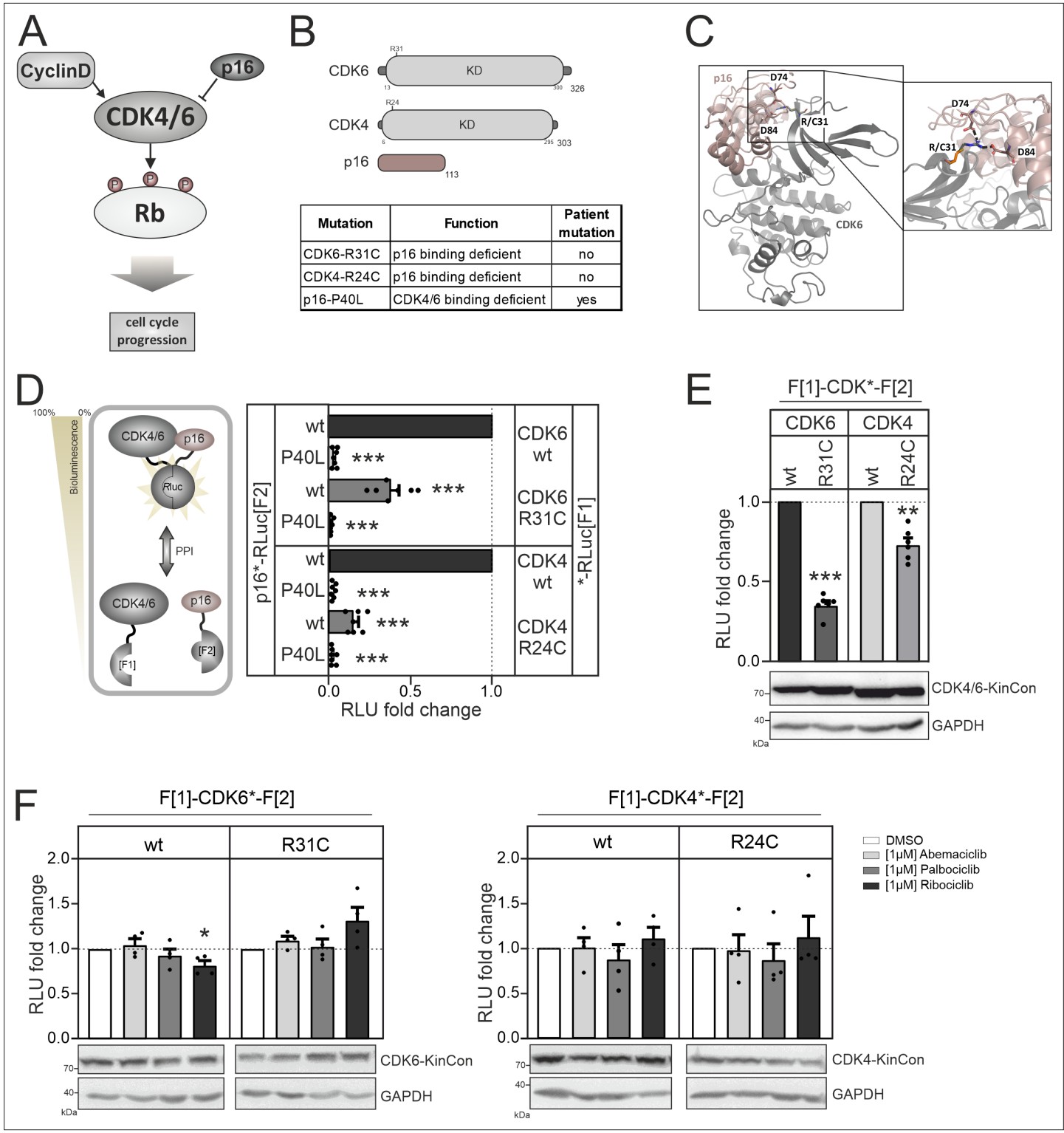

**Figure 4.** CDK4/6 interactions and conformations. (**A**) Illustration of regulatory CDK4/6 interactions and Rb activation. (**B**) Domain organization of CDK4, CDK6, and p16[INK4a]; tested point mutations are listed. (**C**) 3D structure of CDK6 in complex with p16[INK4a]. Crucial amino acids involved in the interaction of the two proteins are highlighted. The R31C mutant is depicted in orange (PDB code: 1BI7, *Russo et al., 1998*). (**D**) Protein:protein interaction (PPI) reporter analyses of the kinases CDK4 and CDK6 with p16[INK4a]. Scheme illustrates CDK4/6 hetero-dimer formation with p16[INK4a] analyzed using a PCA *R*Luc PPI reporter system. PPI induces the complementation of *R*Luc PCA fragments promoting an increase in bioluminescence (HEK293T cells, 48 hr of transient reporter expression). Bars represent the RLU fold change of PPI in relation to wild-type CDK4/6:p16[INK4a] complex (mean ± SEM, n=7 ind. experiments). (**E**) Basal signal of CDK4/6 KinCon reporters with indicated mutations are shown (expressed for 48 hr in HEK293T cells) (mean

*Figure 4 continued on next page*

*Figure 4 continued*

± SEM, n=6 ind. experiments). (**F**) Quantification of alterations of CDK4/6 KinCon reporter bioluminescence signals (HEK293T, expression for 48 hr) upon exposure to indicated CDK4/6i (1 µM) or DMSO for 3 hr (mean ± SEM, n=4 ind. experiments). Statistical significance for D–F: one-sample t-test (*p<0.05, **p<0.01, ***p<0.001).

The online version of this article includes the following source data and figure supplement(s) for figure 4:

**Source data 1.** Raw unedited western blots shown in *Figure 4*.

**Source data 2.** Uncropped and labelled western blots shown in *Figure 4*.

**Figure supplement 1.** Expression of CDK4/6:p16$^{INK4a}$ PPI reporter constructs.

**Figure supplement 1—source data 1.** Raw unedited western blots shown in *Figure 4—figure supplement 1*.

**Figure supplement 1—source data 2.** Uncropped and labelled western blots shown in *Figure 4—figure supplement 1*.

alterations of KinCon activity conformations upon changes of protein interactions and through type I 1/2 and type III inhibitor binding (*Figure 5A–C*), but not type I inhibitors. We provide a summary of the changes in KinCon activity conformations. We have previously shown that for both kinases of the MAPK pathway, MEK1 and BRAF, the respective inhibitors affect primarily the active kinase conformation (*Röck et al., 2019*, *Mayrhofer et al., 2020*, *Fleischmann et al., 2021*, *Fleischmann et al., 2023*). Exemplarily we illustrate that both kinases which are activated through cancer patient mutations (BRAF-V600E and MEK1-K57E) bind the respective kinase inhibitor, resulting in a change in the activity conformations (*Figure 5A and B*). The same observation was made with the MEK1 KinCon upon activation through BRAF-mediated phosphorylation of the reporter (*Fleischmann et al., 2023*). In contrast, it was of interest that for all tested RIPK1 activity conformations we observed that the binding of allosteric RIPK1i promoted a closing of the RIPK1 conformation. In *Figure 5B* we exemplarily illustrate the impact of inhibitor binding on the kinase-dead version RIPK1-K45A. This observation underlines that also inactive RIPK1 complexes are target of drug binding with feasible consequences on kinase scaffolding functions. Kinase activities of CDK4/6 are regulated via defined PPIs (*Nebenfuehr et al., 2020*). We showed that reducing binding affinities for p16$^{INK4a}$ and related inhibitory proteins triggers the opening of the KinCon (*Figure 5C*). However, using the applied standard protocols we have chosen for all CDK4/6 KinCon reporter we have not observed that any of the three clinically applied CDK4/6i tested (such as Abemaciclib) altered the kinase conformation significantly.

In this context we would like to introduce another kinase example displaying regulatory PPI controlling the catalytic kinase protomer. When the regulatory (R) and the catalytic (C/PKAc) subunit interact, the tetrameric PKA holoenzyme complex (R2:C2) is inactive. When second messenger cAMP molecules bind to the R dimer, the complex dissociates and catalytic PKA subunits are activated (*Taylor et al., 2013*). Mutations such as L206R lock the catalytic subunit (PKAc) in its active state. Thus, interactions of regulatory subunits are significantly reduced (*Mayrhofer et al., 2020*, *Bolger, 2022*). Reduction of binding affinities through the use of the general cAMP elevating agent Forskolin triggers conformational changes to the more opened state, mechanistically similar to CDK6-R31C. The PKAc-L206R KinCon mutant already engages this opened activity conformation state (*Figure 5D*). Based on these two examples we assume that binary protein interactions are the prime factors for altering CDK4/6 and PKAc activity conformations.

In the context of kinase compartmentalization and activating kinase mutations we showed that using PKAc and BRAF KinCon reporters as example we could not track alterations of cytoplasmic and nuclear localization (*Figure 5—figure supplement 1*). Furthermore, subcellular localization of PKAc KinCon reporters did not change when the L206R mutation was introduced (*Figure 5—figure supplement 1*). As a control BRAF wt and V600E KinCon reporters were used and no changes in localization were observed.

In this study we focused on a collection of well-studied kinase pathways. In this regard it is important to note that only a limited number of human kinases are targeted by approved drugs. We have illustrated this by using a phylogenetic kinome tree to represent kinase relationships within the human kinome (*Figure 5E*). The blue squares indicate the kinase branches displaying approved kinase blockers. It is evident that numerous protein kinases lack drug candidates. Many of these miss suitable tools/readouts for basic research analyses of physiological and pathological kinase functions. Such kinases have been referred to be part of the so-called 'dark kinome' (*Berginski et al.,*

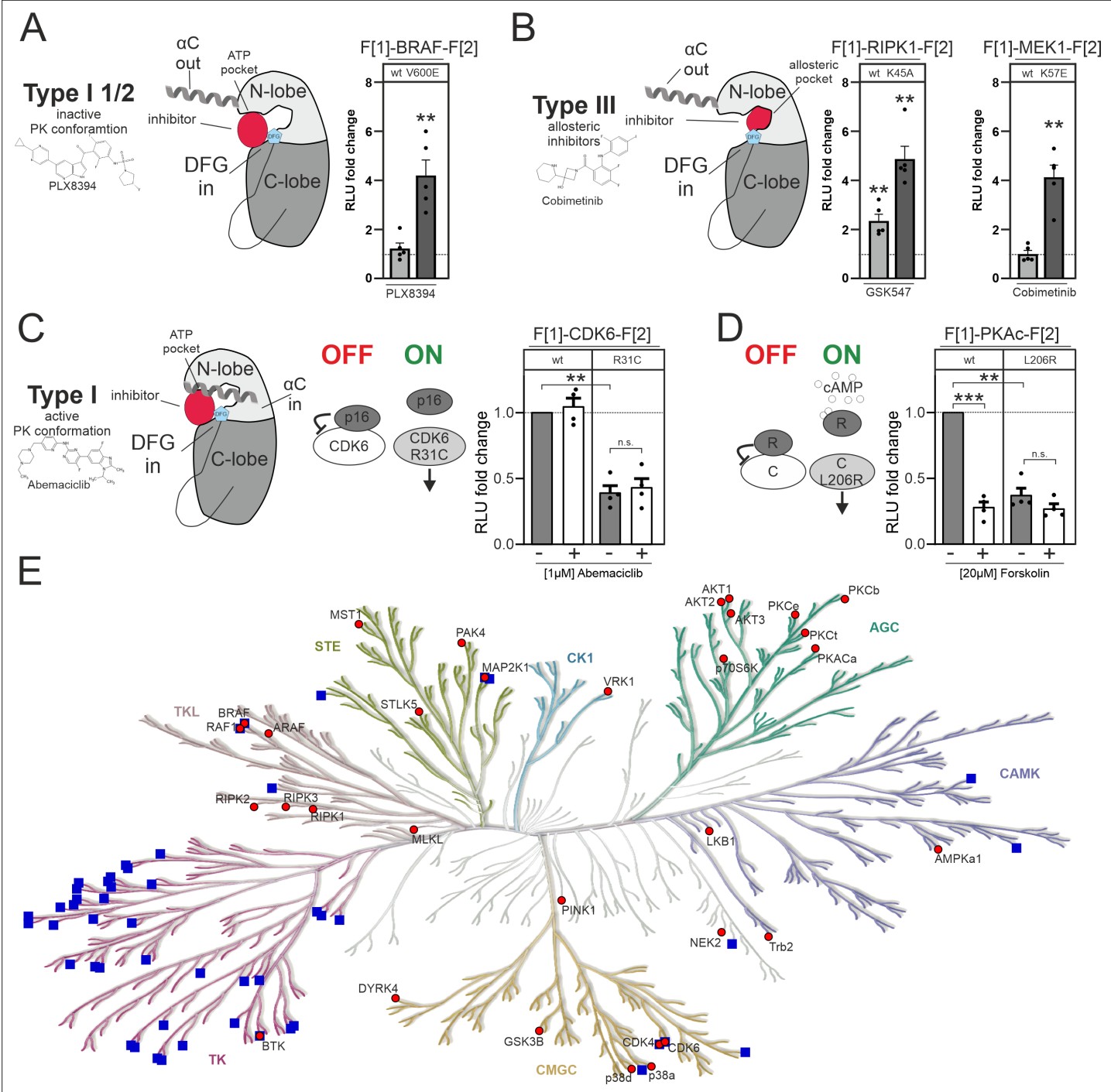

**Figure 5.** Impact of small molecules and protein interactions on kinase activity conformations. (**A+B**) Depiction of molecular interactions of a type I 1/2 and type III kinase inhibitors with a kinase domain (N and C lobe). Impact of PLX8394, Cobimetinib and GSK547 on wild-type (wt) and mutated versions of BRAF, RIPK1, and MEK1 kinase conformation (KinCon) reporters. 48 hr post transfection HEK293T cells expressing respective reporter constructs were treated with indicated inhibitors for 1 hr (1 µM) followed by RLuc PCA analyses (mean ± SEM, n=4/5 ind. experiments). (**C**) Depiction of molecular interactions of a type I kinase inhibitor with a kinase domain (N and C lobe). Impact of Abemaciclib on indicated CDK6 kinase conformations (wt and p16$^{INK4a}$ binding deficient). 48 hr post transfection HEK293T cells expressing respective reporter constructs were treated with indicated inhibitors for 3 hr (1 µM) followed by RLuc PCA analyses (mean ± SEM, n=4 ind. experiments). (**D**) Bioluminescence measurement of PKAc wt and L206R KinCon reporters. HEK293T cells expressing the reporter were treated with 20 µM of Forskolin for 15 min followed by RLuc PCA analyses (mean ± SEM, n=4 ind. experiments). (**E**) Kinome tree displays kinases for which KinCon reporters have been generated (red dots). The blue squares highlight the kinases for

*Figure 5 continued on next page*

*Figure 5 continued*

which approved drugs are available. Generated with https://kinhub.org/kinmap/. Statistical significance for A–D: one-sample t-test (*p<0.05, **p<0.01, ***p<0.001).

The online version of this article includes the following source data and figure supplement(s) for figure 5:

**Figure supplement 1.** Subcellular localization of kinase conformation (KinCon) reporters.

**Figure supplement 1—source data 1.** Raw unedited western blots shown in *Figure 5—figure supplement 1*.

**Figure supplement 1—source data 2.** Uncropped and labelled western blots shown in *Figure 5—figure supplement 1*.

*2021*, *Southekal et al., 2021*). In the kinome tree illustration we have integrated our growing catalog of KinCon reporters (red dots, *Figure 5E*). These conformation reporters represent an extendable toolbox for gaining first and also deeper insights into alterations of kinase activity-related conformation states induced by different means of inhibitors, activators, binders, or mutations. Systematic implementation of this technology in kinase research programs would assist in advancing our understanding of kinase and pseudokinase regulations and functions.

## Discussion

In the field of kinase-related diseases, mutations of different kinds and at different stages of kinase pathways have prompted the development of kinase-specific and small molecule-based inhibitors as primary therapeutic approach (*Ferguson and Gray, 2018*). Despite significant efforts in this direction, only a small fraction of the entire kinome has been successfully targeted by FDA-approved kinase blockers so far (*Lahiry et al., 2010*, *Politi et al., 2015*). Moreover, once a kinase inhibitor is introduced into clinical use, it often encounters the development of drug resistance mechanisms over time (*Longley and Johnston, 2005*). Especially in cancer therapy, these facts pose a significant challenge for identifying and applying efficient small molecule-based therapies. Hence, it is crucial to gain more extensive insights into the cellular drug-target engagement, the molecular consequences of kinase patient mutations, and the evolving patterns of drug resistance mechanisms. For both endeavors new technologies are indispensable for identifying, accurately monitoring, and targeting the dysregulated functions of kinases on the cellular level.

Anticipating and unveiling the cellular mechanism of drug actions may help to develop new treatment concepts with the hope of reducing the diminishing of drug efficacies (*Chan and Ginsburg, 2011*, *Goetz and Schork, 2018*). The KinCon technology, introduced here, seeks to address the previously mentioned challenges. It has the potential to track kinase functions in living cells effectively, providing assistance in measuring difficult-to-assess processes that may include or go beyond phosphotransferase activities. Overall, it offers an innovative solution for understanding kinase activity conformations, which could pave the way for new intervention strategies for kinase entities that have so far not been addressable for drug screenings. So far, this relates to the tracking of kinase scaffold and pseudokinase functions.

Key advantages of the KinCon reporter technology is the robustness of the system to track kinase conformations at varying expression levels. However, in contrast to fluorescence-based reporter read-outs, subcellular analyses and cell sorting are still challenging due to comparably low levels of light emission.

At the beginning of our studies, we underlined the exceptional sensitivity of the KinCon reporter system for tracking kinase activity conformations. We demonstrated that at expression levels far below the endogenous kinase of interest we tracked kinase conformations and their alterations upon drug exposure (*Figure 1E–G*). Tracking enzyme activities at low expression levels is paramount for understanding the details of cellular functions, as many regulatory and spatiotemporal controlled kinase interactions occur under such settings. Such precise measurements are essential for unraveling and targeting pathological kinases functions.

So far, KinCon reporters have been used to assess different conformational states of kinases which are altered by phosphotransferase-activating patient mutations and/or kinase drug binding (*Röck et al., 2019*, *Mayrhofer et al., 2020*, *Fleischmann et al., 2021*, *Fleischmann et al., 2023*). Here, we set out to evaluate consequences of inactivating patient mutations on the complex formation emanating from the kinase LKB1. We aimed to broaden the application spectrum of this reporter

technology for investigating how regulatory protein interactions influence the tumor suppressor functions of the kinase LKB1 (*Partanen et al., 2012*). This holds particular implication for inactivated LKB1, as it suggests that activator compounds could potentially counteract its loss-of-function mutations in cancer. In order to screen or test such reactivator molecules, bioluminescence measurements in living cells would be advantageous, since the complex cellular environment needs to be taken into account. In in vitro assays, which mostly focus on kinase domain activities, regulatory protein interactions are neglected. We have confirmed the notion that the formation of the trimeric complex between STRADα-LKB1-MO25 was necessary for signal propagation to AMPK. In line with this, the LKB1 and STRADα KinCon bioluminescence signals increased upon complex formation and vice versa were reduced back to the baseline signal when a complex breaking mutation in the pseudokinase STRADα was introduced (*Figure 2G*). Further, we showed that tracking structural rearrangements induced by diverse types of mutations can be quantified using KinCon technology (*Figure 2I*). In this setting KinCon measurements of either the kinase (LKB1) or pseudokinase (STRADα) report trimeric complex formation which is pivotal for cytoplasmic kinase pathway activation (*Figure 2E and F*). In contrast to MEK1 and BRAF KinCon reporters, the closed STRADα and LKB1 KinCon conformations seem to represent the active kinase state.

In the next step, we expanded the application of the KinCon reporter to RIPK. Examining the molecular mechanisms that regulate RIPK1 activity poses a challenge because of the intricate nature of the pathway. Functioning as a molecular scaffold, RIPK1 orchestrates signal transduction by coordinating multiple kinases participating in the NFκB pathway (*Moynagh, 2005*, *Meylan et al., 2004*). Moreover, as a regulator of cell death, RIPK1 forms a substantial amyloid-like signalosome in conjunction with RIPK3 and various other factors (*Degterev et al., 2019*). We have shown that indeed pathway activation at different levels of the cascade can be tracked. Further, all tested mutations converted RIPK1 to the more opened conformation (*Figure 3G and H*). Both tested and allosterically acting RIPK1i converted the KinCon reporter back to a more closed conformation (*Figures 3J and 5B*). This consistent impact on the opened kinase conformation indicates a uniform response in altering the enzymes structural state. We assume that this unexpected drug-driven transition of inactive and active RIPK1 complexes to more closed kinase conformation may have relevance for scaffold functions of RIPK1. Additionally, we hypothesize that it could potentially impact the effectiveness of drug binding.

The proto-oncogenic characteristics of CDK4 and CDK6 depend on expression levels and their interactions with a collection of regulatory proteins (*Lin et al., 2001*). We applied the KinCon technology for tracking activity states of both kinases. We demonstrated that regulatory CDK4/6 interactions induce conformational changes that remain unaltered when clinically used type I inhibitors bind to them in the tested cell setting (*Figure 4F*).

In summary, we have extended the KinCon reporter scope of applications. Besides monitoring conformational changes induced by drugs and drug candidates, the reporter system can be applied to accurately track the formation of activated and kinase centered protein complexes. Besides its predictive value in assessing drug effectiveness, the KinCon technology helps consider or identify cellular factors that impact drug candidate binding. KinCon reporters provide unique insights into the molecular dynamics of kinase structure rearrangements. Understanding the molecular motion of kinases as they interact with small molecule inhibitors or regulatory proteins is crucial for designing more effective therapeutic strategies, especially considering that many kinase pathways have so far remained untargeted. The KinCon technology offers a potential avenue to change this.

## Materials and methods

**Key resources table**

| Reagent type (species) or resource | Designation | Source or reference | Identifiers | Additional information |
|---|---|---|---|---|
| Cell line (*Homo sapiens*) | Kidney; Embryo | ATCC | CRL-3216 | HEK293T |
| Cell line (*Homo sapiens*) | Uterus; Cervix, Adenocarcinoma | ATCC | CCL-2 | HeLa |
| Cell line (*Homo sapiens*) | Large intestine; Colon Adenocarcinoma | ATCC | CCL-228 | SW480 |

*Continued on next page*

*Continued*

| Reagent type (species) or resource | Designation | Source or reference | Identifiers | Additional information |
|---|---|---|---|---|
| Transfected construct *human* | MA-T-gRNA-RIP1 | Addgene | 48138 | RIPK1-KO |
| Transfected construct *human* | Cas9-GFP pSp Cas9(BB)–2A-GFP | | | RIPK1-KO |
| Transfected construct *human* | pcDNA3.1-F[1] -LKB1-F[2] | | | LKB1-KinCon |
| Transfected construct *human* | pcDNA3.1-F[1]-LKB1-K78I-F[2] | | | LKB1-KinCon (K78I) |
| Transfected construct *human* | pcDNA3.1-F[1] -LKB1-D176N-F[2] | | | LKB1-KinCon (D176N) |
| Transfected construct *human* | pcDNA3.1-F[1] -LKB1-D194N-F[2] | | | LKB1-KinCon (D194N) |
| Transfected construct *human* | pcDNA3.1-F[1]-LKB1-R74A-F[2] | | | LKB1-KinCon (R74A) |
| Transfected construct *human* | pcDNA3.1-F[1]-LKB1-W308C-F[2] | | | LKB1-KinCon (W308C) |
| Transfected construct *human* | pcDNA3.1-F[1] -LKB1-K48R-F[2] | | | LKB1-KinCon (K48R) |
| Transfected construct *human* | pcDNA3.1-F[1] -STRADα-F[2] | | | STRADα-KinCon |
| Transfected construct *human* | pcDNA3.1-LKB1-3xFlag | | | LKB1-3xFlag |
| Transfected construct *human* | pcDNA3.1-STRADα–3xFlag | | | STRADα–3x Flag |
| Transfected construct *human* | pcDNA3.1-STRADα- H231A/F233A-3xFlag | | | STRADα–3xFlag (H231A / F233A) |
| Transfected construct *human* | pcDNA3.1-F[1]-MLKL-F[2] | | | MLKL-KinCon |
| Transfected construct *human* | pcDNA3.1-F[1]-RIPK1- S14/S15/S166A-F[2] | | | RIPK1-KinCon (S14A/S15A/S166A) |
| Transfected construct *human* | pcDNA3.1-F[1]-RIPK1- S14/S15/S166E-F[2] | | | RIPK1-KinCon (S14E/S15E/S166E) |
| Transfected construct *human* | pcDNA3.1-F[1]-RIPK1-K45A-F[2] | | | RIPK1-KinCon (K45A) |
| Transfected construct *human* | pcDNA3.1-F[1]-RIPK1-D324H-F[2] | | | RIPK1-KinCon (D324H) |
| Transfected construct *human* | pcDNA3.1-F[1]-RIPK1-D324A-F[2] | | | RIPK1-KinCon (D324A) |
| Transfected construct *human* | pcDNA3.1-F[1]-RIPK1-D324E-F[2] | | | RIPK1-KinCon (D324E) |
| Transfected construct *human* | pcDNA3.1-F[1]-RIPK1-C601Y-F[2] | | | RIPK1-KinCon (C601Y) |
| Transfected construct *mouse* | pcDNA3.1-F[1]-CDK6-F[2] | | | CDK6-KinCon |
| Transfected construct *mouse* | pcDNA3.1-F[1]-CDK6-R31C-F[2] | | | CDK6-KinCon (R31C) |
| Transfected construct *mouse* | pcDNA3.1-F[1]-CDK4-F[2] | | | CDK4-KinCon |

*Continued on next page*

*Continued*

| Reagent type (species) or resource | Designation | Source or reference | Identifiers | Additional information |
|---|---|---|---|---|
| Transfected construct *mouse* | pcDNA3.1-CDK6-F[1] | | | CDK6-PPI reporter |
| Transfected construct *mouse* | pcDNA3.1-CDK6-R31C-F[1] | | | CDK6-PPI reporter (R31C) |
| Transfected construct *mouse* | pcDNA3.1-CDK4-F[1] | | | CDK4-PPI reporter |
| Transfected construct *mouse* | pcDNA3.1-CDK4-R24C-F[1] | | | CDK4-PPI reporter (R24C) |
| Transfected construct *mouse* | pcDNA3.1-p16$^{INK4a}$-F[2] | | | p16$^{INK4a}$-PPI reporter |
| Transfected construct *mouse* | pcDNA3.1- p16$^{INK4a}$-P40L-F[2] | | | p16$^{INK4a}$-PPI reporter (P40L) |
| Transfected construct *human* | pcDNA3.1-F[1]-MEK1-F[2] | | | MEK1-KinCon |
| Transfected construct *human* | pcDNA3.1- F[1]-MEK1-K57E-F[2] | | | MEK-KinCon (K57E) |
| Transfected construct *human* | pcDNA3.1-F[1]-PKAc-F[2] | | | PKAc-KinCon |
| Transfected construct *human* | pcDNA3.1-F[1]-PKAc-L206R-F[2] | | | PKAc-KinCon (L206R) |
| Transfected construct *human* | pcDNA3.1-MO25-3xFlag | | | MO25-3xFlag |
| Transfected construct *human* | pcDNA3.1-F[1]-BRAF-F[2] | | | BRAF-KinCon |
| Transfected construct *human* | pcDNA3.1-F[1]-BRAF-V600E-F[2] | | | BRAF-KinCon (V600E) |
| Transfected construct *human* | pcDNA3.1-F[1]-RIPK1-F[2] | | | RIPK1-KinCon |
| Transfected construct *human* | pcDNA3.1-F[1]-RIPK2-F[2] | | | RIPK2-KinCon |
| Transfected construct *human* | pcDNA3.1-F[1]-RIPK3-F[2] | | | RIPK3-KinCon |
| Antibody | GAPDH Rabbit monoclonal | Cell Signaling | 2118 | 1:10000 |
| Antibody | AMPKα Rabbit monoclonal | Cell Signaling | 2532 | 1:1000 |
| Antibody | Phospho-AMPKα(Thr172) Rabbit monoclonal | Cell Signaling | 2535 | 1:1000 |
| Antibody | LKB1 Rabbit monoclonal | Cell Signaling | 3047 | 1:1000 |
| Antibody | Vinculin Rabbit monoclonal | Cell Signaling | 4650 | 1:1000 |
| Antibody | RIP XP Rabbit monoclonal | Cell Signaling | 3493 | 1:1000 |
| Antibody | RIP3 Rabbit monoclonal | Cell Signaling | 13526 | 1:1000 |
| Antibody | CDK6 Mouse monoclonal | Cell Signaling | 3136 | 1:1000 |

*Continued on next page*

*Continued*

| Reagent type (species) or resource | Designation | Source or reference | Identifiers | Additional information |
|---|---|---|---|---|
| Antibody | FLAG-M2 Mouse monoclonal | Sigma-Aldrich | F3165 | 1:1000 |
| Antibody | Renilla Luciferase Rabbit monoclonal | Abcam | Ab185926 [EPR17792] | 1:10000 |
| Antibody | Renilla Luciferase clone 1D5.2 Mouse monoclonal | Millipore | MAB4410 | 1:1000 |
| Drug, chemical compound | Inhibitor | MCE Med Chem Express | HY-50767 | Palbociclib PD 0332991 |
| Drug, chemical compound | Inhibitor | MCE Med Chem Express | HY-16297A | Abemaciclib LY2835219 |
| Drug, chemical compound | Inhibitor | MCE Med Chem Express | HY-15777 | Ribociclib LEE011 |
| Drug, chemical compound | Inhibitor | MCE Med Chem Express | HY-18972 | PLX8394 |
| Drug, chemical compound | Inhibitor | MCE Med Chem Express | HY-114492 | GSK-547 |
| Drug, chemical compound | Inhibitor | MCE Med Chem Express | HY-14622A | Necrostatin 1 S |
| Drug, chemical compound | Inhibitor | MCE Med Chem Express | HY-13064 | Cobimetinib |
| Drug, chemical compound | Inhibitor | MCE Med Chem Express | HY-P7090A | TNFα |
| Drug, chemical compound | Inhibitor | MCE | HY-16658 | z-VAD(OMe)-FMK |

## Expression constructs

All constructs were cloned into the pcDNA3.1 expression vector containing the sequences of Rluc PCA fragments F[1] and F[2] as previously described (*Mayrhofer et al., 2020*, *Fleischmann et al., 2023*, *Röck et al., 2019*). Linear DNA fragments were produced by PCR using Q5 DNA Polymerase. After removing the PCR overhangs with AgeI and HpaI (NEB) respectively and subsequent DNA fragment gel extraction, the PCR insert fragments were isolated using the innuPREP DOUBLEpure Kit (Analytik Jena). The DNA fragments were ligated using T4 DNA ligase (NEB) and amplified using XL10-gold ultracompetent cells. Plasmids were purified by mini- or midiprep (QIAGEN) and verified by Sanger sequencing (Microsynth/Eurofins).

## Cell culture

HEK293T cells were obtained from ATCC (CRL-11268). HEK293T cells were grown in DMEM supplemented with 10% FBS. Transient transfections were performed with Transfectin reagent (Bio-Rad, 1703352). HeLa cells were obtained from ATCC (CCL-2). HeLa cells were grown in DMEM supplemented with 10% FBS. Transient transfections were performed with JetPRIME DNA and siRNA transfection reagent (Polyplus supplied by VWR, 101000046). SW480 cells were obtained from ATCC (CCL-228). SW480 cells were grown in DMEM supplemented with 10% FBS. Transient transfections were performed with JetPRIME DNA and siRNA transfection reagent (Polyplus supplied by VWR, 101000046).

For the HEK293 RIPK1 KO cells HEK293T parental cells were transfected with pMA-T-gRNA-RIP1 targeting (synthetic construct expressing the guide) and Cas9-GFP (pSpCas9(BB)-2A-GFP-Addgene 48138). Both plasmids (one expressing the guide and the other cas9) were transfected with Lipofectamine LTX (Thermo Fisher Scientific). Cells were FACS (BD FACSymphony S6) sorted into single cell clones and analyzed for RIPK1 expression by western blot analysis.

All cells are tested regularly for mycoplasma by PCR using suitable primers and/or Universal Mycoplasma Detection Kit (ATCC, 30-1012K).

## Immunoblotting

Cells were lysed in ice-cold RIPA buffer (50 mM Tris-HCl pH 7.4, 1% NP-40, 0.25% Na-deoxycholate, 120 mM NaCl, 1 mM EDTA, 1 mM PMSF, 1 µg/mL leupeptine/aprotinin/pepstatin, 1 mM $Na_3VO_4$/$Na_4P_2O_7$/NAF) and mixed with 5× Laemmli Buffer. After heating to 95°C for 10 min, the samples were loaded on 10% acrylamide SDS gels for subsequent electrophoresis. Gels were transferred to a PVDF membrane (Roth) using either the Trans-Blot SD Semi-Dry Transfer Cell (Bio-Rad) or the Mini Trans-Blot Cell (Bio-Rad), blocked in TBS-T with 2.5% BSA for 30 min at room temperature and incubated in the primary antibody overnight at 4°C. Blots were incubated with secondary antibodies for 1 hr at room temperature and washed with TBS-T before imaging. Imaging was performed with a FUSION FX (Vilber). Immunoblot images were analyzed using ImageJ (NHI). The signal of the target protein was then normalized on the indicated loading control (e.g. GAPDH or Vinculin). To normalize AMPK phosphory-lation (pAMPK) levels to those of total AMPK, GAPDH-normalized pAMPK levels were divided by their respective GAPDH-normalized total AMPK levels from the same experiment. Primary antibodies used were the rabbit anti-GAPDH (14C10) (Cell Signaling, 2118), rabbit anti-AMPKα (Thr172) (40H9) (Cell Signaling, 2532), rabbit-anti-Phospho-AMPKα (Thr172) (40H9) (Cell Signaling, 2535), rabbit-anti-LKB1 (D60C5) (Cell Signaling, 3047), rabbit-anti-Vinculin (Cell Signaling, 4650), rabbit-anti-RIP (D94C12) XP (Cell Signaling, 3493), rabbit-anti RIP3 (E1Z1D) (Cell Signaling, 13526), anti-CDK6 (DCS83) mouse (Cell Signaling 3136S), mouse-anti-FLAG M2 (Sigma-Aldrich F3165-1MG), rabbit-anti-Renilla Luciferase (EPR17792) (Abcam ab185926), mouse-anti-Renilla Luciferase clone 1D5.2 (Millipore MAB4410).

## Luciferase PPI assay

PCA was performed by growing HEK293T cells in DMEM supplemented with 10% FBS in a 24-well plate format. PCA analyses was performed similarly as previously described in *Röck et al., 2019*. HEK293T cells were grown in DMEM supplemented with 10% FBS. The indicated *R*Luc-tagged constructs, one with *R*Luc-F[1] and one with *R*Luc-F[2], were transiently over-expressed following transfection with TransFectin reagent (Bio-Rad, 1703352) in a 24-well plate format. Fourty-eight hours after transfection the medium was carefully aspirated, the cells were washed once with PBS (1 mM sodium phosphate pH 7.2; 15 mM NaCl) and after addition of 150 µL of PBS to each well, transferred to a 96-well plate (Grainer 96 F-Bottom). Bioluminescence was measured after addition of 20 µL (50 ng) h-coelenter-azine (Nanolight Technology) using the PHERAstar FSX (BMG Labtech) (measurement start time [s]: 0.2; measurement interval time [s]: 10.00; optic module LUM plus; gain: 3600; focal height [mm] 12.5). Data were evaluated using the MARS Data evaluation Software (BMG Labtech).

## Luciferase PCA assay (KinCon assay)

HEK293T cells, cultured in StableCell DMEM (Sigma-Aldrich), were split into 24-well plates at 90.000 cells /well and after 24 hr the indicated plasmids were transfected using TransFectin Lipid reagent (Bio-Rad 1703352) at a total of 50–66 ng/well. After 48 hr of protein expression cells were washed once with PBS (1 mM sodium phosphate pH 7.2; 15 mM NaCl) and after addition of 150 µL of PBS to each well, transferred to a 96-well plate (Grainer 96 F-Bottom). Bioluminescence was measured after addition of 20 µL (50 ng) h-coelenterazine (Nanolight Technology) using the PHERAstar FSX (BMG Labtech) (measurement start time [s]: 0.2; measurement interval time [s]: 10.00; optic module LUM plus; gain: 3600; focal height [mm] 12.5). Data was evaluated using the MARS Data Evaluation Software (BMG Labtech).

## Inhibitors

Inhibitors used were Palbociclib (PD 0332991) (MCE Med Chem Express, HY-50767), Abemac-iclib (LY2835219) (MCE Med Chem Express, HY-16297A), Ribociclib (LEE011) (MCE Med Chem Express, HY-15777), PLX8394 (MCE Med Chem Express, HY-18972), GSK-547 (MCE Med Chem Express, HY-114492), Necrostatin 2 racemate (1S; Nec-1S; 7-Cl-O-Nec1) (MCE Med Chem Express, HY-14622A), Cobimetinib (GDC-0973; XL518) (MCE Med Chem Express HY13064), TNFα (MCE Med Chem Express, HY-P7090A), z-VAD(OMe)-FMK (MCE Med Chem Express, HY-16658), BV6 (MCE Med Chem Express, HY-16701).

## Representation of data

In *Figure 1E and F*, representative experiments of n=4 independent experiments are shown. In these cases absolute bioluminescence values without any normalization are presented. Otherwise, data

was indicated as RLU fold change. Data was normalized on the indicated control condition (through normalization of expression levels determined by the western blotting).

## Statistical analyses

The data were analyzed using GraphPad Prism 8.0. If not indicated otherwise, one-sample t-tests were used to evaluate statistical significance. Values are expressed as the mean ± SEM as indicated. Significance was set at the 95% confidence level and ranked as $*p<0.05$, $**p<0.01$, $***p<0.001$.

## Preparation of structures

The LKB1-STRADα-MO25α (PDB code 2WTK, *Zeqiraj et al., 2009a*) and CDK6-p16$^{INK4a}$ (PDB code: 1BI7, *Russo et al., 1998*) complex structures were prepared using the default setting of the Protein Preparation Wizard *Madhavi Sastry et al., 2013*, *Schrödinger, 2019* in Maestro Schrödinger release 2019-4 (Schrödinger Release 2019-4:*Schrödinger, 2019*). The construct used to generate the LKB1 structure contained the inactivating mutation D194A, intended to prevent $Mg^{2+}$ binding (*Zeqiraj et al., 2009a*). This mutation was converted back to the wt Asp residue using MOE version v2022.02 (Molecular Operating Environment [MOE], 2022.02; Chemical Computing Group ULC, 1010 Sherbrooke St. West, Suite 910, Montreal, QC, Canada, H3A 2R7, 2022). For the calculation of the CDK6 R31C mutant, only the CDK6 residues within 12 Å of p16$^{INK4a}$ were retained. The mutation was then generated using Osprey v2.2beta (*Chen et al., 2009*, *Gainza et al., 2013*) as described before (*Kaserer and Blagg, 2018*). Figures were generated using PyMOL version 2.5.0 (The PyMOL Molecular Graphics System, version 2.5.0 Schrödinger, LLC).

## Acknowledgements

We thank Thomas Nuener for technical, and Erika Lentner and Gabriele Reiter for management support. We thank Alexandra Fritz for her contributions to advance the KinCon technology. Further we would like to thank the Dr. Martin Steinmeyer Foundation for support.

## Additional information

### Competing interests

Philipp Tschaikner, Eduard Stefan: ES and PT are co-founders of KinCon biolabs; KinCon-reporters are subject of patents (WO2018060415A1). The other authors declare that no competing interests exist.

### Funding

| Funder | Grant reference number | Author |
|---|---|---|
| Universität Innsbruck | WS104004 | Selina Schwaighofer |
| Universität Innsbruck | WS104005 | Jakob Fleischmann |
| Horizon 2020 Framework Programme | cONCReTE (872391) | Eduard Stefan |
| Austrian Science Fund | 10.55776/P30441 | Eduard Stefan |
| Austrian Science Fund | 10.55776/P32960 | Eduard Stefan |
| Austrian Science Fund | 10.55776/P35159 | Eduard Stefan |
| Austrian Science Fund | 10.55776/I5406 | Eduard Stefan |
| Österreichische Forschungsförderungsgesellschaft | 877163 | Eduard Stefan |

The funders had no role in study design, data collection and interpretation, or the decision to submit the work for publication.

## Author contributions
Valentina Kugler, Data curation, Formal analysis, Validation, Investigation, Visualization, Writing – original draft; Selina Schwaighofer, Data curation, Formal analysis, Funding acquisition, Validation, Investigation, Visualization, Writing – original draft; Andreas Feichtner, Florian Enzler, Jakob Fleischmann, Sophie Strich, Data curation, Formal analysis, Validation, Investigation; Sarah Schwarz, Visualization; Rebecca Wilson, Resources, Data curation; Philipp Tschaikner, Conceptualization, Writing – review and editing; Jakob Troppmair, Veronika Sexl, Pascal Meier, Conceptualization; Teresa Kaserer, Conceptualization, Data curation, Software, Visualization, Writing – review and editing; Eduard Stefan, Conceptualization, Resources, Supervision, Funding acquisition, Methodology, Writing – original draft, Project administration, Writing – review and editing

## Author ORCIDs
Valentina Kugler  http://orcid.org/0009-0001-1238-9530
Selina Schwaighofer  https://orcid.org/0009-0001-3273-7715
Pascal Meier  http://orcid.org/0000-0003-2760-6523
Teresa Kaserer  https://orcid.org/0000-0003-0372-1885
Eduard Stefan  https://orcid.org/0000-0003-3650-4713

Reviewer #1 (Public review): https://doi.org/10.7554/eLife.94755.3.sa1
Reviewer #2 (Public review): https://doi.org/10.7554/eLife.94755.3.sa2
Author response https://doi.org/10.7554/eLife.94755.3.sa3

---

# Additional files

## Supplementary files
• MDAR checklist

## Data availability
Data supporting the findings of this study are available within the article and the corresponding raw data is freely available as a PDF-File on Zenodo (https://doi.org/10.5281/zenodo.12073536).

The following dataset was generated:

| Author(s) | Year | Dataset title | Dataset URL | Database and Identifier |
|---|---|---|---|---|
| Kugler V, Schwaighofer S, Feichtner A, Enzler F, Fleischmann J, Strich S, Schwarz S, Wilson R, Tschaikner P, Troppmair J, Sexl V, Meier P, Kaserer T, Stefan E | 2024 | Kinases in motion: impact of protein and small molecule interactions on kinase conformations | https://zenodo.org/records/12073536 | Zenodo, 10.5281/zenodo.12073536 |

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
