## [Editor Report · eLife assessment]

This article reports an **important** bioluminescence-based reporter system to evaluate kinase conformations. This assay is applied to four different kinases that have unique, very special regulatory features, thereby indicating that the assay can be used to provide **convincing** evidence on the conformational state of a large number of kinases. This paper will be of interest to researchers working on kinases and their conformational states.

---

## [Referee Report · Reviewer #1 (Public review)]

Summary:

This technical report by Kugler at al., expands the application of a fluorescence-based reporter to study the conformational state of various kinases. This reporter, named KinCon (Kinase Conformation), interrogates the conformational state of a kinase (i.e., active vs. inactive) based on engineering complementary fusion proteins that fluoresce upon interaction. This assay has several advantages as it allows studying full-length kinases, that is, the kinase domain and regulatory domains, inside the cell and under various experimental conditions such as the presence of inhibitors or activator proteins, and in wildtype and mutants involved in disease states.

Strengths:

One major strength of this study is that it is quite comprehensive. The authors use KinCon for four different kinases, BRAF, LKB1, RIP and CDK4/6. These kinases have very different regulatory elements and associated proteins, which the authors explore to study their conformational state. Moreover, they use small molecule inhibitors or mutations to further dissect how the conformational state of the kinase in disease states. The collective set of results strongly suggests that KinCon is a versatile tool that can be used to study many kinases of biomedical and fundamental importance. Given that kinases are extensively studied by researchers in academia or industry, KinCon could have a broad impact as well.

Weaknesses:

This manuscript, however, also has several weaknesses that I outline below. These weaknesses decrease the overall level of impact on the manuscript, as is.

• The manuscript is exceedingly long. For instance, the introduction provides background information for each kinase that is further expanded in the results section. I think the background information for each kinase in the Introduction and Results sections can be significantly reduced to highlight the major points. Otherwise, not only does the manuscript become too long, but also the main points get diluted.

• Similarly, the figure legends are very long, providing information that is already in the main text or in Methods. The authors should provide the essential information to understand the figure.

• A major concern throughout the manuscript is the use of the word "dynamics," which is used in the text in various contexts. The authors should clarify what they understand for dynamics of conformation. Are they measuring how the time-dependent process by which the kinase is interconverting between active and inactive states? It seems to me that the assays in this report evaluate a population of kinases that are in an open or close conformation (i.e., a particular state in each experimental condition) but there is not direct information how the kinase goes from one state to the other. In that sense, the use of dynamics is unclear. Also, the use of dynamics in different sentences in ambiguous. Here are a few examples but this should be revised throughout the manuscript:

- Line 27: dynamics of full-length protein kinases. Is this referred to dynamics of conformational interconversion between inactive and active states?

- Line 138: dynamic functioning of kinases. No clear what that means.

- Line 276: ... alters KinCon dynamics. Not clear if they are measuring time-dependent process or a single point.

- Figure legend 4F: dynamics of CDK4/6 reporters. Again, not clear how the assay is measuring dynamics.

Nonetheless, in my opinion the authors use proper terminology that describes their assay in which the term dynamics is not used: Title (... impact of protein and small molecule interactions on kinase conformations) and Line 89 (... reporter can be used to track conformational changes of kinases...)

• The authors use the phrase that KinCon has predictive capabilities (abstract and line 142). What do the authors refer to this?

• The authors indicate that KinCon is a highly sensitive assay. Can the authors elaborate on what high sensitivity means? For example, can they discuss how other fluorescence-based approaches that are less sensitive would not be able to accomplish the same type of results or derive similar conclusions? Can they provide a resolution metric both in space and time? Given that the authors state that this is a technical report, this information is of relevance.

• The authors nicely describe how KinCon works in Figure 1B and part of 1C. I do think that the bottom of panel 1C needs to be revised, as well as the text describing the potential scenarios of potency, efficacy and synergism.

- One issue with this part of Figure 1C is that it is not clear what the x-axis in the 3 plots refer to. Is this time? Is this concentration of a small molecule, inhibitor or binding partner? This was confusing also in the context of the term dynamics used throughout the text. The terms potency, efficacy and synergism should be subtitles or the panels and the x-axis should be better defined, especially for a non-specialized reader.

- Related to this part of Figure 1C is the text. The authors mention potency, effectiveness and synergy (Line 195). Can the authors use more fundamental terminology related to these three scenarios, for example, changes in activation constant, percent of protein activates? Also, why synergy is only related to effectiveness? Can synergy also be associated to potency?

- Lastly, the use of these three cartoons gives the impression that the experimental results to come will follow a similar representation. Instead, the results are presented in bar plots for many different conditions. I think this will lead to confusion for a broad audience.

• For a non-expert reader, can the authors clarify the use of tracking basal conformations vs. transient over-expression of the various KinCon constructs? Moreover, the authors use the term transient over-expression for 10, 16, 24 and 48 h (Line 203). This, to a non-expert reader, seems not transient.

• Regarding Figure 1E and similar graphical representations: Why is the signal (RLU) non-linear with time? If the fluorescence of the KinCon construct is linearly related with its expression or concentration inside the cell, one would expect a linear increase. Have the authors plotted RLU/Expression band intensity to account for changes in protein concentration? For instance, some of the results within Figure 3 are normalized to concentration on the reporter expression level.

• For the results with LKB1, the authors claim that intermediate fold change in fluorescence (Figure 2E) is due to a partially closed intermediate state (Line 262). Can the authors discard the possibility by which there is a change in populations of active and inactive that on average give intermediate values?

• The authors claim in Line 274 that mutations located at the interface of the LKB1/STRADalkpha complex affect interactions and hypothesize that allosteric communication between LKB1 and STRADalpha is essential for function. Given that this mutations are at the interaction interface, why would the authors postulate an allosteric mechanism that evokes an effect distant to the interaction/active site? Could it be that function requires surface contacts alone that are disrupted by the mutations?

• I was unable to find text to explain the following: Figure 2I shows the mutation R74A as n.s., but in the text only W308C is mentioned to not change fluorescence. Could the authors clarify why R74A is not discussed in the text? Maybe this reviewer missed the text in which it was discussed. Similarly, the author states in line 326 that the study included an analysis of RIPK2. However, I was unable to find results, graphs or additional text discussing RIPK2.

• Some figures of RLU use absolute values, percentages and fold change. Is there a reason why the authors use different Y-axis values? These should be explained and justified in Methods. Similarly, bars for wt in Figures 3D, G, or 4D, E,F show no errors. How are the authors normalizing the data and repeats so that there is no error, and are they treating the rest of the data (i.e., mutants and/or treated with small molecules) in the same way?

• Lastly, the section starting in Line 472 reads more like a discussion of results from different type of inhibitors used in this study that results on its own. The authors should consider a new subtitle as results or make this section a discussion.

---

## [Referee Report · Reviewer #2 (Public review)]

Summary:

Protein kinases have been very successfully targeted with small molecules for several decades, with many compounds (including clinical drugs) bringing about conformational changes that are also relevant to broader interactions with the cellular signaling networks that they control. The authors set out to develop a targeted biosensor approach to evaluate distinct kinase conformations in cells for multiple kinases in the context of incoming signals, other proteins and small molecule binding, with a broad goal of using the KinCon assay to confirm (and perhaps predict) how drug binding or signal perception changes conformations and outputs in the presence of cellular complexes; this work will likely impact on the field with cellular reporters of kinase conformations a useful addition to the toolbox.

Strengths:

The KinCon reporter platform has previously been validated for well-known kinases; in this study, the team evaluate how to employ a full-length kinase (often containing a known pathological mutation). The sensitive detection method is based on a Renilla luciferase (RLuc)protein fragment complementation assay, where individual RLuc fragments are present at the N and the C terminus of the kinase. This report, which is both technical and practical in nature, co-expresses the kinase with known interactors (at low levels) in a high throughput format and then performs pharmacological evaluation with known small molecule kinase modulators. This is explained nicely in Figure 1, as are the signaling pathways that are being evaluated. Data demonstrate that V600E BRAF iexposed to vemurafenib is converted to the inactive conformation, as expected. In contrast, the more closed STRADα and LKB1 KinCon conformations appear to represent the more active state of the complexed kinase, and a W308C mutation (evaluated alongside others) reverses this effect. The authors then evaluated necroptotic signaling in the context of RIPK1/3 under conditions where RIPK1 and RIPK3 are active, confirming that the reporters highlight the active states of both kinases. Exposure to compounds that are known to engage with the RIPK1 arm of the pathway induce bioluminescence changes consistent with the opening (inactivation) of the kinase. Finally, the authors move to an important drug target for which clinical drugs have arrived relatively recently; the CDK4/6 complexes. These are of additional importance because kinase-independent functions also exist for CDK6, and the effects of drugs in cells usually relies on a downstream marker, rather than demonstration of direct protein complex engagement. The data presented are interpreted as the formation of complexes with the CDK inhibitor p16INK4a; reducing the affinity of the interaction through mutations drives an inactive conformation, whilst the application of CDK4/6 inhibitors does not, implying binding to the active conformation.

Weaknesses:

(1) The work is very solid, and uses examples from the literature and also extends into new experimental space. An obvious weakness is mentioned by the authors for the CKDK data, in that measurements with Cyclin D (the activating subunit) are not characterised, although Cyclin D might be assumed to be present?

(2) The work with the trimeric LKB1 complex involves pseudokinase, STRADalpha, whose conformation is also examined as a function of LKB1 status; since STRAD is an activator of LKB1, a future goal should be the evaluation of the complex in the presence of STRAD inhibitory/activating small molecules.

---

## [Author Response]

The following is the authors’ response to the original reviews.

We would like to thank you and the two Reviewers for the thoughtful evaluation of the manuscript and the support for publication. We have addressed all points raised by the two Reviewers.

- We have extensively streamlined the manuscript. Repetitive passages regarding the respective kinase cascades have been removed.

- We improved the presentation of the main Figures (mainly labeling and font size):

- Figure 1: C, D, E, F o Figure 2: C, E, F, G, I, o Figure 3: D o Figure 4: F

- Figure 5: A, B, C, D, E

- We integrated new SI-data related to kinase functions, expression and the ‘cell-type comparisons’ of the KinCon reporter system (Figure Supplement 4, 5).

Below you will find a detailed point-by-point response.

**Reviewer #1 (Recommendations For The Authors):**
Regarding the issue of the use of the word "dynamics," as described in the public review, here are a few examples of ambiguous use in different sentences: o Line 27: dynamics of full-length protein kinases. Is this referring to the dynamics of conformational interconversion between inactive and active states?- Line 138: dynamic functioning of kinases. It is not clear what this means. o Line 276: ... alters KinCon dynamics. Not clear if they are measuring time-dependent process or a single point.- Figure legend 4F: dynamics of CDK4/6 reporters. Again, not clear how the assay is measuring dynamics.In my opinion, the authors use proper terminology that describes their assay in which the term dynamics is not used: Title: "... impact of protein and small molecule interactions on kinase conformations" and Line 89 "... reporter can be used to track conformational changes of kinases...".

We have replaced the “dynamics” sections.

- Line 27: The understanding of the structural dynamics of…

- Line 91: This reporter can be used to track dynamic changes of kinases conformations…

- Line 139: Conventional methods often fall short in capturing the dynamics of kinases within their native cellular environments…

- Line 146: Such insights into the molecular structure dynamics of kinases in intact cells…

- Line 199: In order to enhance our understanding of kinase structure dynamics…

- Line 276: These findings underline that indeed the trimeric complex formation alters….

- Figure Legend 4F: Quantification of alterations of CDK4/6 KinCon reporter bioluminescence signals…

The authors state that KinCon has predictive capabilities (abstract and line 142). What do the authors mean by this?

Previously we have benchmarked the suitability of the KinCon reporter for target engagement assays of wt and mutated kinase activities. With this we determined specificities of melanoma drugs for mutated BRAF variants (Mayrhofer 2020, PNAS).

The authors indicate that KinCon is a highly sensitive assay. Can the authors elaborate on what high sensitivity means?

With sensitivity we mean that we can detect conformation dynamics of the reporter at low expression levels of the hybrid protein expressed in the cell line of choice.

- Line 209: Immunoblotting of cell lysates following luminescence measurements showed expression levels of the reporters in the range and below the endogenous expressed kinases (Figure 1E). …

- Line 219: Using this readout, we showed that at expression levels of the BRAF KinCon reporter below the immunoblotting detection limit, one hour of drug exposure exclusively converted BRAF-V600E to the more closed conformation (Figure 1F, G, Figure Supplement 1B).

- Line 221: These data underline that at expression levels far below the endogenous kinase, protein activity conformations can be tracked in intact cells. …

For example, can they discuss how other fluorescence-based approaches that are less sensitive would not be able to accomplish the same type of results or derive similar conclusions? Can they provide a resolution metric both in space and time? Given that the authors state that this is a technical report, this information is of relevance.

We highlight the key pros & cons of the KinCon reporter technology in following sections:

-Line 529: The KinCon technology, introduced here, seeks to address the previously mentioned challenges. It has the potential to become a valuable asset for tracking kinase functions in living cells which are hard to measure solely via phosphotransferase activities. Overall, it offers an innovative solution for understanding kinase activity conformations, which could pave the way for more novel intervention strategies for kinase entities with limited pharmaceutical targeting potential. So far, this relates to the tracking of kinase-scaffold and pseudo-kinase functions.

- Line 535: Key advantages of the KinCon reporter technology is the robustness of the system to track kinase conformations at varying expression levels. However, in contrast to fluorescence-based reporter read-outs subcellular analysis and cell sorting are still challenging due to comparable low levels of light emission

The authors nicely describe how KinCon works in Figure 1B and part of 1C. I do think that the bottom of panel 1C needs to be revised, as well as the text describing the potential scenarios of potency, efficacy, and synergism.One issue with this part of Figure 1C is that it is not clear what the x-axis in the 3 plots refers to. Is this time? Is this concentration of a small molecule, inhibitor, or binding partner? This was confusing also in the context of the term dynamics used throughout the text. The terms potency, efficacy, and synergism should be subtitles, or the panels and the x-axis should be better defined, especially for a non-specialized reader.Related to this part of Figure 1C is the text. The authors mention potency, effectiveness, and synergy (Line 195). Can the authors use more fundamental terminology related to these three scenarios, for example, changes in activation constant, and percent of protein activates? Also, why synergy is only related to effectiveness? Can synergy also be associated with potency?

Thank you for bringing this up, we have revised Figure 1C to better reflect the mentioned effects of potency. To avoid confusion, we removed the illustration for drug synergism. Accordingly, we have integrated the axis descriptions for the presented dose-response curves.

Thus, we have further streamlined the text in the introduction – examples are shown below:

- Line 195: Light recordings and subsequent calculations of time-dependent dosage variations of bioluminescence signatures of parallel implemented KinCon configurations aid in establishing dose-response curves. These curves are used for discerning pharmacological characteristics such as drug potency, effectiveness of drug candidates, and potential drug synergies (Figure 1C)

- Figure 1C: Shown is the workflow for the KinCon reporter construct engineering and analyses using KinCon technology. The kinase gene of interest is inserted into the multiple cloning site of a mammalian expression vector which is flanked by respective PCA fragments (-F[1], -F[2]) and separated with interjacent flexible linkers. Expression of the genetically encoded reporter in indicated multi-well formats allows to vary expression levels and define a coherent drug treatment plan. Moreover, it is possible to alter the kinase sequence (mutations) or to co-express or knock-down the respective endogenous kinase, interlinked kinases or proteinogenic regulators of the respective pathway. After systematic administration of pathway modulating drugs or drug candidates, analyses of KinCon structure dynamics may reveal alterations in potency, efficacy, and potential synergistic effects of the tested bioactive small molecules (schematic dose response curves are depicted)

Lastly, the use of these three cartoons gives the impression that the experimental results to come will follow a similar representation. Instead, the results are presented in bar plots for many different conditions. I think this will lead to confusion for a broad audience.

The bottom panel of Figure 1C is not the depiction of real experiments but rather an illustration of fitted dose-response curves. We would like to present previous demonstrations of doseresponse curves using BRAF KinCon data and ERK phosphorylation (Röck 2019, Sci. Advances)

We further agree with the reviewer and have therefore added a new part in the methods section addressing the evaluation of data extensively.

- Line 668: In Figure 1 E and F, a representative experiment of n=4 independent experiments is shown. In these cases, absolute bioluminescence values without any normalization are shown. Otherwise, data was indicated as RLU (relative light unit) fold change. This means the data was normalized on the indicated control condition either with normalization of the western blot or without; as indicated.

For a non-expert reader, can the authors clarify the use of tracking basal conformations vs. transient over-expression of the various KinCon constructs? Moreover, the authors use the term transient over-expression for 10, 16, 24, and 48 h (Line 203). This, to a non-expert reader, does not seem transient.

We have revised the manuscript to clarify it:

- Line 207: We showed that transient over-expression of these KinCon reporters for a time frame of 10h, 16h, 24h or 48h in HEK293T cells delivers consistently increasing signals for all KinCon reporters (Figure 1E, Figure Supplement 1A).

- Figure 1E Representative KinCon experiments of time-dependent expressions of indicated KinCon reporter constructs in HEK293T cells are shown (mean ± SEM). Indicated KinCon reporters were transiently over-expressed in 24-well format in HEK293T cells for 10h, 16h, 24h and 48h each.

Regarding Figure 1E and similar graphical representations: Why is the signal (RLU) nonlinear with time? If the fluorescence of the KinCon construct is linearly related to its expression or concentration inside the cell, one would expect a linear increase. Have the authors plotted RLU/Expression band intensity to account for changes in protein concentration? For instance, some of the results within Figure 3 are normalized to concentration on reporter expression level.

Out intention was to show that varying expression levels can be used for the illustrated target engagement assays.Indeed, the represented elevations of RLU might be due to factors such as:

- Doubling times of cells

- Cell density

- Media composition (which changes over time)

- Reporter protein stabilities

- Abundance of interactors of kinases

For the results with LKB1, the authors claim that intermediate fold change in fluorescence (Figure 2E) is due to a partially closed intermediate state (Line 262). Can the authors discard the possibility by which there is a change in populations of active and inactive that on average give intermediate values?

Based on our experience with KinCon reporter conformation states of kinases we tested so far, we assume that the presented data reflects an intermediate state. We agree that it needs further validation. We have changed the text accordingly:

- Line 264: Upon interaction with LKB1 this conformation shifts to a partially closed intermediate state.

The authors claim in Line 274 that mutations located at the interface of the LKB1/STRADalpha complex affect interactions and hypothesize that allosteric communication between LKB1 and STRADalpha is essential for function. Given that these mutations are at the interaction interface, why would the authors postulate an allosteric mechanism that evokes an effect distant from the interaction/active site? Could it be that function requires surface contacts alone that are disrupted by the mutations?

We agree with the reviewer and changed our argumentation for this point:

- Line 276: These findings underline that indeed the trimeric complex formation alters the opening and closing of the tested full-length kinase structures using the applied KinCon reporter read out

I was unable to find text to explain the following: Figure 2I shows the mutation R74A as n.s., but in the text, only W308C is mentioned to not change fluorescence. Could the authors clarify why R74A is not discussed in the text? Maybe this reviewer missed the text in which it was discussed.

We adapted the manuscript and include the R74A mutation as followed:

- Line 296: Among these mutations, only the W308C and R74A mutation prevented significant closing of the LKB1 conformation when co-expressed with STRADα and MO25 (Figure 2I).

In Figure 2I where the individual measurements of the LKB1-R74A KinCon are highlighted in red to better emphasize the deviations. In the case of the R74A mutation the effect seen might be due to the high deviation between the experiments (Highlighted in red). These deviations are much higher when compared to either the wt or the W308 mutant, and can also be seen in the LKB1-R74A-KinCon only condition (white). Even though no significant closing of the LKB1 conformation could be observed in the case of R74A, we believe, since the trend of the conformation closing upon complex formation is still visible that the effect is still there. Further replicates would be necessary to validate this theory.

Similarly, the authors state in line 326 that the study included an analysis of RIPK2. However, I was unable to find results, graphs, or additional text discussing RIPK2.

The RIPK2 conformation was analyzed in Figure 3C (page 12).

Some figures of RLU use absolute values, percentages, and fold change. Is there are reason why the authors use different Y-axis values? These should be explained and justified in Methods. Similarly, bars for wt in Figures 3D, G, or 4D, E, F show no errors. How are the authors normalizing the data and repeats so that there is no error, and are they treating the rest of the data (i.e., mutants and/or treated with small molecules) in the same way?

We have changed the Y-axis values. Now, throughout the manuscript we show that there is a RLU fold-change. Except are selected experiments when solely absolute RLU values are shown (such as Figure 1E, F). We have also decided to integrate a paragraph into the methods section (Line 655). Figure 3D was changed as well.

- Line 668: In Figure 1 E and F, a representative experiment of n=4 independent experiments is shown. In these cases absolute bioluminescence values without any normalisation are shown. Otherwise, data was indicated as RLU fold change. This means the data was normalized on the indicated control condition (either with normalization of the western blot or without; as indicated).

The data is generally normalized on wt or untreated conditions, when the cells were treated with small molecules for target engagement assays.

Lastly, the section starting in Line 472 reads more like a discussion of results from different types of inhibitors used in this study that results on its own. The authors should consider a new subtitle such as results or make this section a discussion.

We agree with the reviewer and this part of the results was split into a new section of the result:

- Line 455: “Effect of different kinase inhibitor types on the KinCon reporter system”.

**Reviewer #2 (Recommendations For The Authors):**
I have a few suggestions, since the paper is a distillation of a vast amount of work and tells a useful story.(1) The work is very solid, uses examples from the literature, and also extends into new experimental space. An obvious weakness is mentioned by the authors for the CKD data, in that measurements with Cyclin D (the activating subunit) are not characterized, although Cyclin D might be assumed to be present.

We performed experiments with the CDK4/6 KinCon reporters and co-expressed CyclinD with a ratio of 1:3 (HEK293T cells, expression for 48h). However, in the context of inhibitor treatments we could not track conformation changes in these initial experiments. The cells were treated with the indicated CDK4/6i [1µM] for 3h. This seems to not impact the conformation of CDK4/6 wt or mutated KinCon reporters. There is a tendency that CyclinD co-expression promotes CDK4/6 conformation opening (data not shown).

**Author response image 1. sa3fig1:** Bioluminescence signal of CDK4/6 KinCon reporters with co-expressed CyclinD3 (HEK293T, expression for 48h) upon exposure to indicated CDK4/6i [1µM] or DMSO for 3h (mean ± SEM, n=3 ind. experiments). No significant changes using the current setting.

(2) The work with the trimeric LKB1 complex involves pseudokinase, STRADalpha, whose conformation is also examined as a function of LKB1 status; since STRAD is an activator of LKB1. A future goal should be the evaluation of the complex in the presence of STRAD inhibitory/activating small molecules.

Thank you for this great idea, we are currently compiling a FWF grant application to get support for such a R&D project.

Minor points• Have any of the data been repeated in a different cell background? This came to mind because HeLa cells lack LKB1, which might be a useful place to test the LKB1 data in a different context.

This experiment was performed and we show it in Figure Supplement 5. Further, we followed the advice of the reviewer and performed suggested experiments. We integrated the colon cancer cell line SW480 into the experimental setup. Overall, three cell settings showed the same pattern of KinCon reporter analyses for LKB1-STRADα-MO25 complex formation utilizing the LKB1- and STRADα-KinCon reporters.

• The study picks up the PKA Cushings Syndrome field, which makes sense, and data are presented for L206R. PMID 35830806 explains how different patient mutations drive different signaling outcomes through distinct complex formations, and it would be interesting to discuss how mutations in KinCon complexes, especially those with mutations, could affect sub-cellular localization. Could the authors explain if this was done for any of the proteins, whose low experimental expression is a clear advantage, but is presumably hard to maintain across experiments?

The feedback of the reviewer motivated us to perform subcellular fractionation experiments. They were performed with PKAc wt and L206R KinCon reporters as well as BRAF wt and V600E reporters. We were not able to see major differences between the wt and mutated reporter constructs in respect to their nucleus: cytoplasm localizations (Figure Supplement 4). For your information, in a R+D project with the mitochondrial kinase PINK1 we see localization of the reporter as expected almost exclusively at the mitochondria fraction.

- Line 495: In this context of activating kinase mutations we showed that using PKAc (wt and L206R) and BRAF (wt and V600E) reporters as example we could not track alterations of cytoplasmic and nuclear localization (Figure Supplement 4). Furthermore, subcellular localization of PKAc KinCon reporters did not change when L206R mutant was introduced (Figure Supplement 4). As a control BRAF wt and V600E KinCon reporters were used and also no changes in localization was observed.

• I suggest changing PMs (Figure 2 and others) simply to mutation, I read this as plasma membrane constantly.

We agree and we have changed it to “patient mutation” in Figure 2C, Figure 3E, Figure 4B.